# High-dimensional super-resolution imaging reveals heterogeneity and dynamics of subcellular lipid membranes

Karl Zhanghao [1,2,6 ✉], Wenhui Liu[3,6], Meiqi Li [2,6], Zihan Wu[1], Xiao Wang[4], Xingye Chen[3], Chunyan Shan [4], Haoqian Wang [3], Xiaowei Chen[4], Qionghai Dai[3], Peng Xi [1,2 ✉] & Dayong Jin [1,5 ✉]

Lipid membranes are found in most intracellular organelles, and their heterogeneities play an essential role in regulating the organelles' biochemical functionalities. Here we report a Spectrum and Polarization Optical Tomography (SPOT) technique to study the subcellular lipidomics in live cells. Simply using one dye that universally stains the lipid membranes, SPOT can simultaneously resolve the membrane morphology, polarity, and phase from the three optical-dimensions of intensity, spectrum, and polarization, respectively. These high-throughput optical properties reveal lipid heterogeneities of ten subcellular compartments, at different developmental stages, and even within the same organelle. Furthermore, we obtain real-time monitoring of the multi-organelle interactive activities of cell division and successfully reveal their sophisticated lipid dynamics during the plasma membrane separation, tunneling nanotubules formation, and mitochondrial cristae dissociation. This work suggests research frontiers in correlating single-cell super-resolution lipidomics with multiplexed imaging of organelle interactome.

[1] UTS-SUStech Joint Research Centre for Biomedical Materials & Devices, Department of Biomedical Engineering, College of Engineering, Southern University of Science and Technology, Shenzhen Guangdong, P.R. China. [2] Department of Biomedical Engineering, College of Engineering, Peking University, 100871 Beijing, China. [3] Department of Automation, Tsinghua University, 100084 Beijing, China. [4] State Key Laboratory of Biomembrane and Membrane Biotechnology, College of Life Sciences, Peking University, 100871 Beijing, China. [5] Institute for Biomedical Materials & Devices (IBMD), University of Technology Sydney, Sydney NSW 2007, Australia. [6] These authors contributed equally: Karl Zhanghao, Wenhui Liu, Meiqi Li. ✉email: karl.hao.zhang@gmail.com; xipeng@pku.edu.cn; jindy@sustech.edu.cn

Lipid membranes surround most subcellular compartments and play a significant role in the sophisticated machinery of cells[1–3]. They are heterogeneous across compartments and are dynamically regulated during the cell cycle[4]. Their morphology, composition, and phase synergistically regulate biophysical membrane properties, membrane protein functionalities, and lipid–protein interactions[2], which assists the coordination of organelles. Despite the significant role of lipid membranes, approaches to study them in vivo have been limited, as the available spatiotemporal resolution, throughput, and stability over the long-term observation are far from satisfactory.

Here we employ Nile Red[5], a common intracellular lipid dye, to stain lipid membranes universally in live cells (Fig. 1). From the GFP colocalization images and the Nile Red membrane morphology images, we identify that Nile Red successfully labels at least ten subcellular compartments. Figure 1b shows that the mitochondria-GFP, Golgi-GFP, ER-GFP, lysosome-GFP, early endosome-GFP, and late endosome-GFP colocalize the corresponding organelles. Since Nile Red emits green-yellow fluorescence only in lipid droplets, colocalization of Nile Red signals in the green and yellow channels indicates the organelle type of lipid droplets (Fig. 1c). The plasma membrane, nuclear membrane, and tunneling nanotube can be easily identified from their morphology in the intensity image of Nile Red (Fig. 1d–f).

Two distinct physical properties of Nile Red are first explored in our work: the emission spectrum changes are measured by two-color ratiometric imaging (Fig. 2b), and the wobbling dynamics of fluorescent dipoles are resolved by polarization modulation (Fig. 2c). The emission spectrum of Nile Red will red shift in polar environment and blue shift in nonpolar environment[6]. Membranes in mammalian cells consist of three lipid categories: glycerophospholipid, sphingolipid, and cholesterol (Fig. 2d, e)[2]. The lipid polarity descends from unsaturated glycerophospholipid, saturated glycerophospholipid, sphingolipid, to cholesterol, so that the emission spectrum of Nile Red will shift from red to blue accordingly. The sphingolipid-rich membrane tends to form the solid-like phase, the membrane rich of cholesterol and saturated glycerophospholipid tends to form the liquid-ordered (Lo) phase, and the membrane rich of unsaturated glycerophospholipid tends to form the liquid-disordered (Ld) phase[1,7]. Polarization imaging of fluorophores can reveal the orientational wobbling of molecular dipoles to distinguish ordered or disordered lipid phases[8–11]. In the ordered membrane, the wobbling of dipoles is more restricted so that Nile Red exhibits stronger polarization response with increased modulation depth. In disordered membrane, the wobbling of dipoles is more flexible so that Nile Red exhibits weaker polarization response with decreased polarization modulation depth.

Besides these multiple responsive optical properties, the fluorogenic dye Nile Red is highly bio-compatible[5,12] and only emits fluorescence within the hydrophobic environment of lipid membranes, which is highly suitable for long-term monitoring of lipid dynamics in living cells. All these above-mentioned merits of Nile Red will benefit the in vivo studies of lipid membranes, once the organelle-level imaging resolution can be achieved.

## Results

**Super-resolution imaging with spectrum and polarization optical tomography**. Without sufficient resolution, as shown in Fig. 2a, the crowded compartments in the Nile Red-stained U2-OS cells induce messy signals all over the imaging volume, which deteriorates both the ultrastructure and the functional measurements. To overcome this challenge, we first apply structured illumination microscopy (SIM) to improve the spatial resolution by two times from the blurry wide field (WF) image (Fig. 2a). We demonstrate that the improved resolution also significantly lowers the measurement errors of the lipid polarity and phase by simulating two nearby molecules with different emission spectrum and polarization response (Fig. 2f, g). Nevertheless, SIM improves only the spatial imaging and hyperspectral detection, but not the polarization imaging due to the missing cross harmonics, based on our previous work of polarized SIM[13].

Therefore, we have further developed a spectrum and polarization optical tomography (SPOT) method (details in Supplementary Fig. 1 and Supplementary Note) to increase the measuring accuracies for both lipid polarity and phase (Supplementary Figs. 2–4). The principle of SPOT is to obtain optical sectioning with structured illumination[14,15] with two or more phase shifts. Using the HiLo method[15] by fusing the High and Low spatial frequency information for each pattern direction, the less modulated out-of-focus background is rejected from the in-focus signals. Similar to polarized SIM[13], three pattern directions are applied for polarization modulation analysis, so that SPOT requires six images with structured illumination, less than the typical nine images required by SIM. Besides, SPOT can also be applied to the SIM dataset and take advantage of the doubled spatial resolution. The attenuated out-of-focus background significantly increases the measurement accuracies in both emission ratio and polarization modulation depth. We verified the SPOT method by labeling the actin filaments of Nile Red-stained U2-OS cells with fluorophores whose emission spectrum and polarization response are known. The experimental results demonstrate the superior performance of SPOT, compared with those of either WF or SIM. The measurement accuracies have been increased by SPOT from 71 to 95% in the emission ratio compared with WF, from 62 to 91% in the modulation depth (Supplementary Fig. 3), and from 87 to 96% in the dipole orientation compared with WF and SIM (Supplementary Fig. 4).

**High-dimensional polarity map and phase map**. The membranes with different lipid properties guarantee the incompatible biochemical processes of different compartments and participate in the compartment interactions. We first characterized the different emission ratios and modulation depths of different lipid membranes, which are shown in the polarity map of Fig. 3a and the phase map of Fig. 3b with appreciable contrasts. Furthermore, we quantified the lipid heterogeneity across colocalized compartments of the plasma membrane, nuclear membrane, endoplasmic reticulum (ER), mitochondria, lipid droplet, Golgi apparatus, lysosome, and endosome (Fig. 3c, d). Nuclear membrane and ER show no significant difference in both lipid polarity and phase. They have the highest emission ratio and relative high modulation depth. Plasma membrane and early endosome share low emission ratio and highest modulation depth. Lipid droplets have the lowest emission ratio and also the lowest modulation depth. We find that the trends in the lipid measurement, especially the measured emission ratio (lipid polarity), were highly consistent with the lipid compositions of different compartments (Supplementary Fig. 5). These compartments can be further categorized into six groups on the polarity-phase plot based on their similarities and differences (Fig. 3e). These high-dimensional super resolution maps and statistical analysis suggest rich biophysical information behind the lipid composition and organization of each subcellular compartment.

**Intra-organelle lipid heterogeneity**. The super-resolution mapping power offered by SPOT modality can further reveal the lipid heterogeneity within the same organelle, such as mitochondria

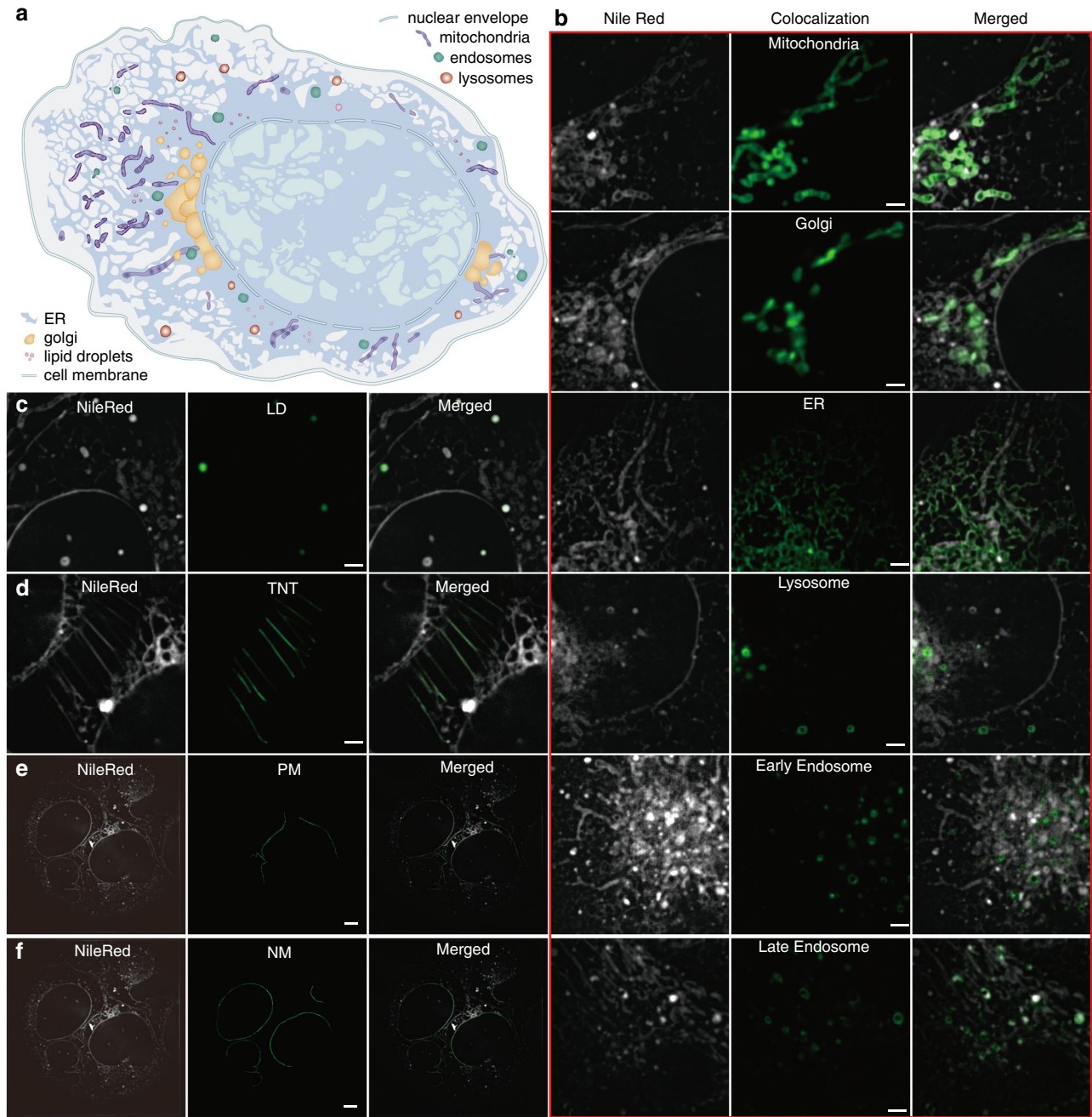

**Fig. 1 Colocalization of Nile Red-stained compartments in U2-OS cells with spectrum and polarization optical tomography (SPOT) intensity images.**
**a** The cartoon diagram of various compartments in the cell. **b** GFP colocalization of mitochondria, Golgi apparatus, ER, lysosome, early endosome, and late endosome. The images in each left column are fluorescent images of Nile Red in the yellow channel (excitation: 561 nm, emission: 578–614 nm). The images in the middle are the fluorescent images of GFP in the green channel (excitation: 488 nm, emission: 505–545 nm). **c** Colocalization of Nile Red signals in both green and yellow emission channels, when excited by the 488-nm laser, identifies lipid droplets, as Nile Red emits both green and yellow fluorescence only in lipid droplets. **d–f** The tunneling nanotube (TNT), plasma membrane (PM), and nuclear membrane (NM) are identified from their morphologies, which are manually marked in the middle column. All the experimental results in **b–f** were repeated at least three times with independently prepared samples. Scale bar: (**b–d**) 2 μm; (**e–f**) 5 μm.

and lipid droplets. We observed the lipid heterogeneity on the outer membrane and the cristae of mitochondria directly from the functional images, and the statistics show a significantly higher lipid polarity in the cristae than the outer membrane (Fig. 3f, g). This observation can be further confirmed in the mitochondria images in Fig. 4 and Supplementary Movie. The rise in the lipid polarity may be correlated to the cholesterol change from 7.1% in the outer membrane to 2.3% in the cristae[16]. The lipid heterogeneity is also found in the large and grown lipid droplet, where both the emission ratio and the modulation depth are smaller in the core than in the shell (Supplementary Figs. 2 and 6). These observations are highly consistent with the fact that the core of lipid droplet stores low-polarity neutral lipids and is coated by a monolayer phospholipid shell[17,18].

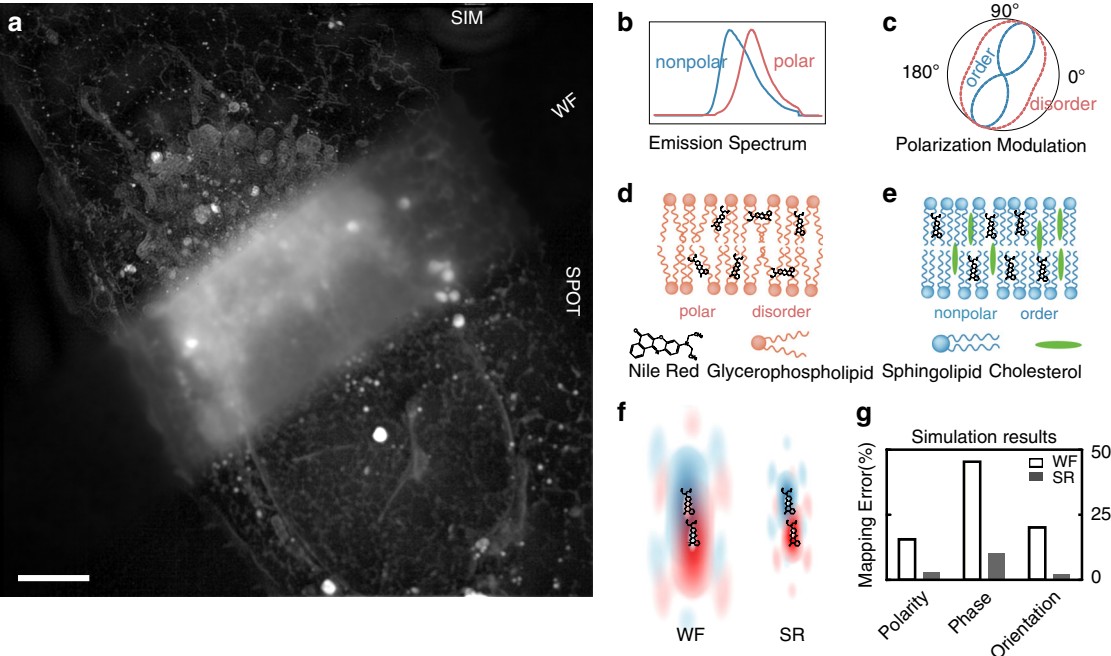

**Fig. 2 Principle of the high-dimensional super-resolution imaging of subcellular lipid membranes. a** The max intensity projection image (40 × 40 × 2.75 μm³) shows a U2-OS cell stained by Nile Red that labels lipid membranes of various compartments. In contrast to the blurry wide field (WF) image, 3D structured illumination microscopy (SIM), and SPOT clearly resolve the subcellular structures. Similar results were repeated five times independently. **b** The schematic emission spectrum of Nile Red is shifted by the polarity of its environment, i.e., a blue-shift in nonpolar lipids and a red-shift in polar lipids. **c** When Nile Red inserts into the lipid membrane, the wobbling behavior of the fluorescent dipole reflects the lipid phase, which can be quantified by polarization modulation depth. In the ordered phase, the dipole orientation is more uniform, leading to a higher modulation depth (the schematic curve in blue); while in the disordered phase, the dipole orientation is more anomalous, resulting in a smaller modulation depth (the schematic curve in red). **d**, **e** The enormous types of lipids can be categorized into glycerophospholipid, sphingolipid, and cholesterol. The polarity descends from glycerophospholipid, sphingolipid, to cholesterol. Glycerophospholipid alone tends to form an disordered phase of membranes, while cholesterol and sphingolipid assist the formation of an ordered phase. **f**, **g** Improved spatial resolution can attenuate influencing signals from other molecules when measuring the target molecule. By simulating two molecules with different emission spectrum and dipole behavior, the errors can be reduced from 15.1% to 2.2% in polarity, from 45.5% to 7.9% in phase, and from 19.1% to 1.6% in dipole orientation with doubled lateral and axial resolution. The influence of noise on measurement accuracies is illustrated with simulations in Supplementary Fig. 8. In Nile Red-stained U2-OS cells, the abundant out-of-focus signal lowers the measuring accuracy, where SPOT is demonstrated with superior performance to WF and SIM with comprehensive experimental results (Supplementary Figs. 3 and 4). Scale bar: 5 μm.

**Dynamic lipid properties of endosomes and lipid droplets**. We further revealed that the lipid properties of endosomes and lipid droplets are highly dynamic at different developmental stages. By colocalizing early endosomes with Rab5a-GFP and late endosomes with Rab7a-GFP, we observed the increase in polarity and the decrease in phase from early endosomes to late endosomes (Fig. 3h). Furthermore, early endosome and plasma membrane, late endosome and lysosome show no significant difference in lipid properties. These results reflect the maturation process of endosomes. Early endosomes are formed during the endocytosis process, whose lipid membrane originates from plasma membranes. During maturation, late endosomes have reduced proportion of cholesterol and sphingolipid[19] (Fig. 3i) and develop into lysosomes. Moreover, we observed that the emission ratio dropped when the size of lipid droplets grew, suggesting more neutral lipids are stored in lipid droplets and lower the polarity of lipid membranes during the maturation (Supplementary Fig. 6).

**Monitoring lipid rearrangements during cell division**. By recording the last-stage division process of two U2-OS cells for 10 min at the non-bleaching speed of 0.33 Hz (Fig. 4f), we observed at least three exciting phenomena, such as the separation of plasma membranes, the formation of tunneling nanotubules (TNT), and the cristae dissociation of mitochondria

(Fig. 4a). Before the conjunct plasma membrane divides into two separate membranes, we observed that the lipid polarity dropped, and the lipid order rose while the lipid properties of a non-dividing plasma membrane keep unchanged (Fig. 4b). After the division completed, both the lipid polarity and phase recovered. These results suggest that lipid remodeling is required for the dividing of plasma membranes. About 2 min after the division of the plasma membrane, we captured the formation of several TNTs[20] (Fig. 4c, d and Supplementary Movie 1). While the TNTs seemed to be stretched from the plasma membrane, the lipid polarity, and phase of these TNTs were quite different from the plasma membrane and even from one to another. The emission ratio and modulation depth displayed a large deviation between TNTs, while the lipid properties of each single TNT remained little change over time. These results suggest that each TNT may be responsible for a specific task during cell-to-cell interaction, for example, transporting a specific type of organelles. We also observed the cristae dissociation of mitochondria during the process (Fig. 4e). We found that the lipid polarity and phase rose during the cristae dissociation and dropped back in the time window of ~2 min while the lipid polarity of the control mitochondria remained constant. Together with the lipid heterogeneity in the outer membrane and the cristae shown in Fig. 3f, g, these results further suggest that the dissociated lipid in the

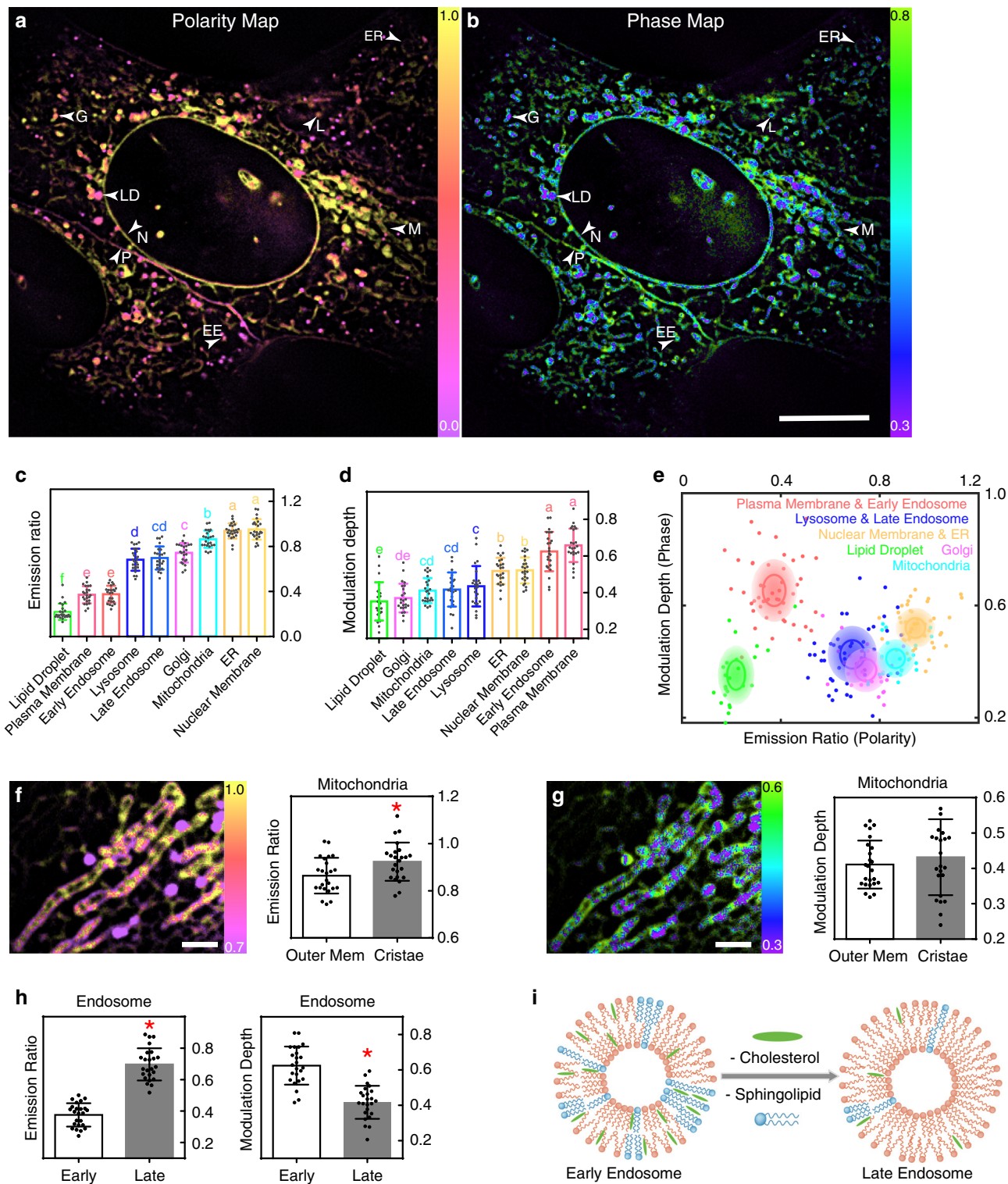

cristae would slowly pass the outer membrane and be released into the cytoplasm. Dramatic membrane rearrangements occur during cell division[4,21,22], and the high spatiotemporal monitoring of lipid properties by SPOT can provide insights on how the lipid is regulated in cytokinesis.

## Discussion
Previous studies only measured either the lipid polarity[19,23,24] or the lipid phase[8–11]. Some of these studies assumed that a higher lipid polarity led to a lower degree in lipid order or the other way around. This assumption is correct in the synthesized vesicles with one or two types of lipids. However, lipid diversity is much more complex in mammalian cells that produce more than 10,000 different types of lipids. Besides, the lipid phase is strongly associated with the lipid–lipid interactions[2], so that it should not be solely determined by the lipid polarity. Using SPOT, we presented the capability of correlative imaging of both the emission ratio and modulation depth, to quantify the complementary

**Fig. 3 Heterogeneity analysis of subcellular lipid membranes. a, b** The polarity map and the phase map obtained by SPOT show noticeable contrasts between the lipid membranes of different compartments. The lipid polarity is quantified by the emission ratio and warm-color coded, while the lipid phase is resolved by the polarization modulation depth and cool-color coded. Similar results were repeated three times independently. **c–e** The statistics ($n = 24$ for each kind of organelle) measured on each colocalized compartment further quantify their lipid heterogeneity. Duncan's multiple range test is used to analyze the significance of differences between multiple groups where the threshold of $p$ is set to 0.05. The histogram of the emission ratio (**c**) shows the heterogeneous lipid polarity, and the histogram of the modulation depth (**d**) shows the heterogeneous lipid phase ($n = 24$ for each kind of organelle), where characters on the bars indicate significant differences. The polarity-phase plot (**e**) further categorizes the six groups of the compartments according to their similarities and differences, in which the solid line and the transparent ellipse show the standard deviation $\sigma$ and $2\sigma$ of the measurement. **f, g** The polarity map and the phase map of mitochondria reveal the heterogeneity between the outer membrane and the cristae. The statistics ($n = 24$ mitochondria) also show a significantly higher lipid polarity in the outer mitochondria membrane but no significant difference in the lipid phase. Two-sided $t$-test is applied and $p = 0.0118, 0.4265$ for **f, g** respectively. **h, i** The statistics ($n = 24$ early endosomes and late endosomes) measured with colocalization show the increase in polarity and the decrease in phase from early endosomes to late endosomes. Two-sided $t$-test is applied and $p < 0.0001$ in both polarity and phase. The red 'asterisk' in **f** and **h** indicate that there exist significant differences. All the data in the bar charts are presented as mean values ± SD. The arrows in **a, b** indicate the possible compartments recognized by their morphology and lipid properties. The statistical results are based on ≥3 independent experiments. Source data are provided as a Source Data file. EE early endosome, G Golgi apparatus, L lysosome or late endosome, LD lipid droplet, M mitochondria, N nuclear membrane, P plasma membrane. Scale bar: **a, b** 10 μm; **f, g** 2 μm.

information of lipid polarity and phase. In our results, as the lipid polarity drops, the lipid phase increases in mitochondria, late endosome, lysosome, early endosome, and plasma membrane, which is consistent with the above assumption. The lower emission ratio suggests a higher proportion of sphingolipid or cholesterol that results in a higher lipid order. However, lipid droplet serves as a counterexample that shows both low lipid polarity and order. This is because the core of lipid droplets stores the low-polarity lipid, such as triglyceride in disorder. Nuclear membrane and ER have high lipid polarity, but with a relatively high ordered phase, which may be related to the sophisticated lipid-lipid interactions. The wide existence of lipid heterogeneity uncovered by SPOT demonstrates its complementary capacity to mass spectroscopy[25,26] in quantifying the molecular composition of lipids, as mass spectroscopy is insufficient for live-cell studies.

Compared with the other existing fluorescence polarization microscopy techniques, SPOT is superior in optical throughput that correlatively obtains the high-dimensional information from six raw images within tens of milliseconds. In contrast, the typical acquisition time for polarization modulation by point-scanning confocal imaging is in the range of seconds to minutes[10]. Polarization modulation by spinning disk confocal imaging allows parallel acquisitions that significantly increase the imaging speed to ~10 frames per second[27], but it is still slower than SPOT and has not become available with commercial systems. Polarization demodulation can also obtain super resolution with deconvolution in spatioangular space[28–30]. However, this technique requires a sparse distribution of dipole orientations that is not the case in cells with crowded membranous structures. Single molecule localization microscopy (SMLM) is capable of measuring the orientation and wobbling of individual dipoles[31–35], and Nile Red, when using at ultra-low concentrations, is also compatible with SMLM[36]. However, a higher concentration of Nile Red is required to stain all types of membranes, which rules out the use of SMLM imaging.

Besides the optical sectioning in the axial dimension, the lateral resolution remains diffraction-limited for SPOT, while the resolution of SPOT-SIM3D is doubled. As a result, the polarity measurement is the averaged value of the lipid assembly within the point spread function (PSF). When curved membranous structures are within the PSF, the measurement will result in a significantly lower polarization modulation depth due to an ensemble dipole of the different molecular dipole orientations. For example, the membranous structure of Golgi apparatus is crowded and cannot be clearly resolved by SPOT, which further leads to a reduced lipid order in the measurement of Golgi apparatus.

Another limitation lies in the correlation between the measured optical properties and the lipid properties of the membrane, which is related to the dye used. Though we have carefully calibrated the emission ratio of Nile Red in different solvents with known polarity (Supplementary Fig. 7), the environment of cellular lipid membranes is much more sophisticated. Therefore, the penetration depth and the orientation of the dye may differ in the membranes of different lipid compositions, which could influence its emission ratios[37]. Fortunately, the optical properties of different lipid membranes are simultaneously measured with the same dye Nile Red and under the same condition, which minimizes these influencing factors. Future work better includes characterizations of the dyes in in-vitro lipid vesicles with known lipid composition, which can provide more insights on this discussion. In cells, the spectrum shift of Nile Red is mostly attributed to the polarity of lipid, because the fluorescence of Nile Red is negligible in water and the emission spectrum of lipid dyes is little affected by lipid surface modification protein such as lipoprotein[38].

Most microscopes can image four or fewer color channels, the number of which is far less than the types of intracellular organelles. The lipid heterogeneity analysis offered by the polarity-phase plot of SPOT imaging opens the opportunities for organelle identification and segmentation. The synergistic use of membrane morphology, lipid polarity, and lipid phase can classify more than 10 types of organelles. While plasma membrane, nuclear membrane, TNT, ER, and mitochondria with cristae structures are easily identified from their morphologies, other round-shaped organelles are inferred from their lipid properties. It should be noted that sometimes misclassifications between Golgi and late endosomes may occur due to the overlapping lipid properties (Fig. 3e). Artificial intelligence[39] can be further employed to integrate the intensity, polarity, and phase images for the automatic segmentation of the compartments in the future. While significant efforts have been made in multiplexed imaging, including spectrum unmixing[40], optical barcoding[41], and lifetime tuning[42,43], our method based on the heterogeneity features of lipid membranes in the spectrum and polarization domain, has extended the new dimensions with much simplified and practical implementation potentials. This work, therefore, suggests a avenue in long-term inspections of live-cell lipidomics and organelle interactome at high spatiotemporal resolution.

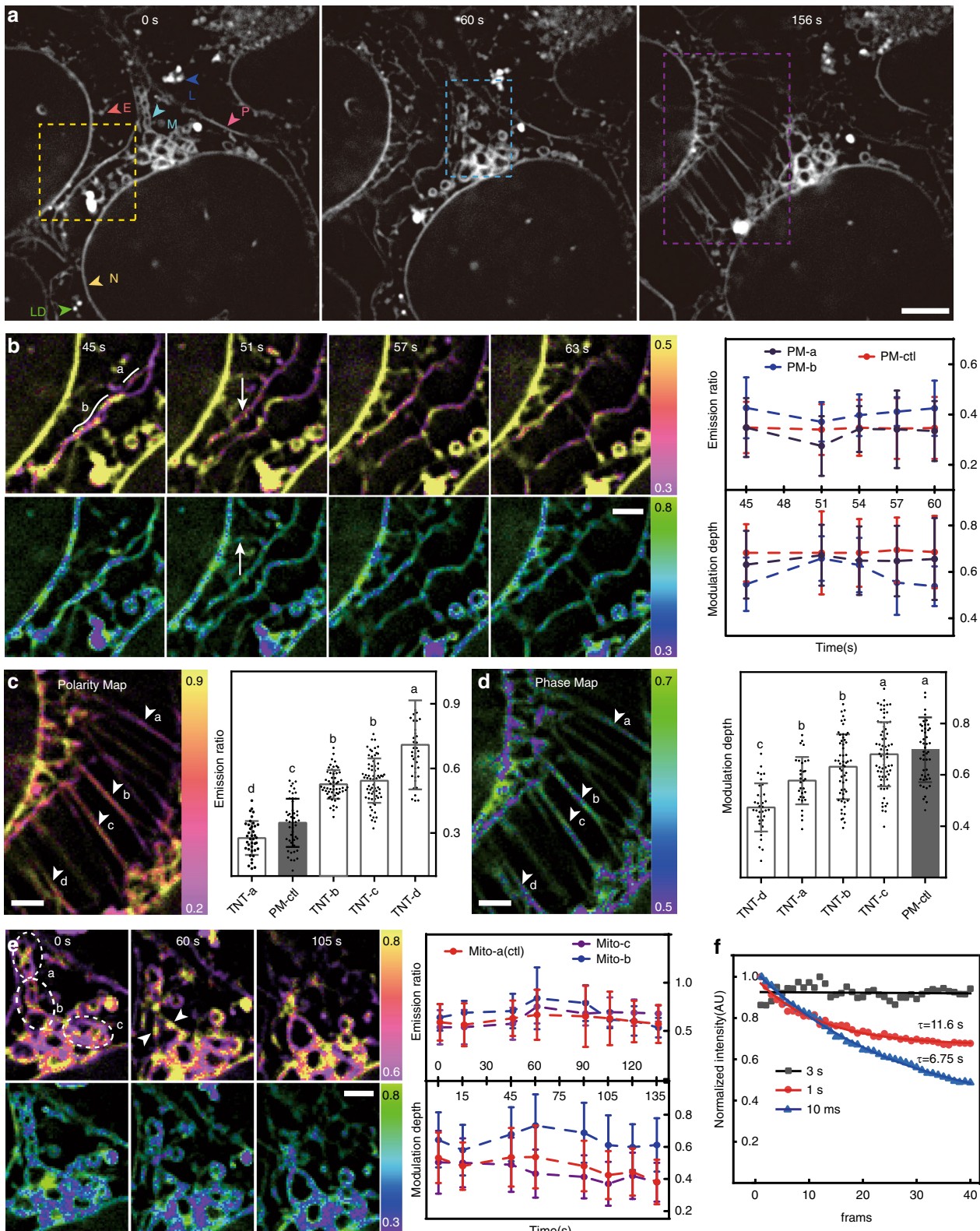

## Methods

**Sample preparation.** Human osteosarcoma U2-OS cell lines (HTB-96, ATCC, USA) were cultured in Dulbecco's Modified Eagle's medium (DMEM, GIBCO, USA) containing 10% heat-inactivated fetal bovine serum (FBS, GIBCO, USA) and 100 U/ml penicillin and 100 µg/ml streptomycin solution (PS, GIBCO, USA) at 37 °C in an incubator with 95% humidity and 5% $CO_2$. For the living cell imaging, the cells were plated at the desired density on the µ-Slide 8 Well (80827, ibidi, USA)

and 1 µg/ml Nile Red (N1142, Invitrogen, USA) was added into the culture medium 1 h before imaging and was present during imaging. For colocalization experiments, the cells were transfected 16 h before imaging with the plasmids of early endosome-GFP (Rab5a, C10586, BacMam 2.0, CellLight, USA), late endosome-GFP (Rab7a, C10588), ER-GFP(ER signal sequence of calreticulin and KDEL, C10590), Golgi-GFP (Golgi-resident enzyme N-acetylgalactosaminyl-transferase 2, C10592), Lysosome-GFP (Lamp1, C10596), Mitochondria-GFP

**Fig. 4 Long-term monitoring of the lipid dynamics of multiple compartments. a** The time-lapse images recorded the late-stage division of two U2-OS cells at 3 s acquisition interval for 10 mins. The bleaching analysis in **f** shows the non-bleaching imaging power at this interval. Similar results were repeated in two independent experiments. **b** The enlarged images of the yellow box show the lipid polarity and phase during the division of the plasma membrane. The emission ratio drops during the division and recovers after the separation of two plasma membranes, while the modulation depth first rises and then drops. During the same period, the control plasma membrane that is not dividing keeps constant in both polarity and phase ($n = 22, 26, 27$ measurement pixels for PM-a, PM-b, PM-ctl respectively). **c, d** The enlarged images of the purple box show the formation of TNTs after the separation of plasma membranes. The measurements reveal a large variation in the lipid polarity and phase among these TNTs. The histogram plots the pixel values on each TNT, and the characters on the bars indicate significant differences of each group ($n = 39, 56, 67, 37, 47$ measurement pixels for TNT-a, TNT-b, TNT-c, TNT-d, PM-ctl respectively). **e** The enlarged images of the blue box show the cristae dissociation of mitochondria during the process. The curves of Mito-b and Mito-c show an increase in the outer membrane in the emission ratio, which drops back afterward. In contrast, the lipid polarity of Mito-1 without cristae dissociation is more constant ($n = 95, 107, 88$ measurement pixels for Mito-a, Mito-b, Mito-c respectively at 0 s). Data in all the line and bar charts in **b**–**e** are presented as mean values ± SD. **f** The curves show the average fluorescence signal imaged with different acquisition interval and are fitted with an exponential function to calculate the halftime ($n = 3$ independent experiments). The exposure time of each image is 40 ms, and the total acquisition time of a SPOT dataset (six raw images) is 240 ms. Similar results are observed in $\geq 2$ experiments. Source data are provided as a Source Data file. Scale bar: **a** 5 μm; **b**–**g** 2 μm.

(leader sequence of E1 alpha pyruvate dehydrogenase, C10600), and incubated overnight. In total, 1 μg/ml Nile Red was added into the culture medium 1 h before imaging.

Fixed cells were plated on #1.5H cover glasses (CG15XH, Thorlabs, USA) and fixed with 4% formaldehyde (R37814, Invitrogen, USA) for 15 min. After washing with PBS, Alexa Flour 568 Phalloidin (A12380, Invitrogen, USA) was used to stain the actin filaments for 1 h at room temperature and washed with PBS. Then the coverslip was sealed onto the cavity of the slide (MS15C1, Thorlabs, USA) with the Nile Red solution in PBS inside. No permeabilization reagents such as Triton X-100 or antifade reagents were used during the whole process. The illumination calibration beads (F8887, 580/605, 0.1 μm, Invitrogen, USA) were diluted at proper concentration and then coated on the coverslips treated with the Poly-L-lysine solution (P8920, Sigma, USA). The coverslip was sealed on the slide with the prolong diamond antifade mountant (P36965, Invitrogen, USA).

**System setup and image acquisition.** The home-built SIM system is based on the SLM-SIM setup described before[13]. Two linearly polarized continuous-wave lasers, 488 nm (Coherent, Sapphire LP 488-200 mW) and 561 nm (CNI, MGL-FN-561 nm-200 mW), were used and switched by an acoustic-optic tunable filter (AOTF, AA Opto-Electronic, France). A ferroelectric liquid-crystal spatial light modulator (SLM; Forth Dimension Displays, SXGA-3DM-DEV) was used to generate the illumination stripes. Only the ±1 diffracted beams can pass the spatial mask. A vortex half-wave plate (VHWP, Thorlabs, WPV10L-532) was applied to modulate the polarization of the ±1 beams to be s-polarized. Two identical dichroic mirrors (Chroma, ZT405/488/561/640rpcv2) were placed perpendicular to each other to compensate for the polarization distortion. A 4f system was used to relay ±1 order beams to the back focal plane of the objective (Nikon, CFI SR HP Apochromat TIRF ×100 oil, NA 1.49). The emitted fluorescence was filtered by the emission filter (EM, Chroma, ZET405/488/561/640mv2) and afterward separated into four channels by the four-way image splitter system (CAIRN, MultiSplit V2). Three dichroic mirrors (DM, Chroma, ZT488rdc/ZT561rdc/ZT640rdc) were installed in the MultiSplit system to separate the four channels (blue: 417–471 nm, green: 505–545 nm, yellow: 578–614 nm, red: 658–749 nm). The four detection channels were projected onto different areas of an sCMOS camera (Hamamatsu, ORCA-Flash4.0 V2). The SLM, DAQ, and sCMOS were synchronized by a DAQ (National Instruments, USB6363) as illustrated in a previous work[44,45]. The whole system was controlled by the Micro-manager software.

For SPOT imaging with only Nile Red staining, only the 561-nm laser was used for excitation, and only the yellow/red detection channel was used. Six patterns (three directions multiplied by two phases with π phase shift) were applied to the sample sequentially. For SPOT imaging with GFP colocalization, the sample was excited by the 488-nm laser and 561-nm laser sequentially, and the green/yellow/red detection channel was used. The commercial OMX-SIM system (DeltaVision OMX SR, GE, USA) was used to perform 3D-SIM imaging. Two 3D-SIM datasets were acquired sequentially. The first one used the 561 excitation channel and 561 detection channel; the second one used the 561 excitation channel and the 640 detection channel.

Two calibration experiments were performed before each experiment. The calibration of the illumination nonuniformity was performed with a slide of 100 nm fluorescent beads (F8887, Invitrogen, USA), which was described as elsewhere[13]. The other calibration experiment was to register different detection channels. The alignment slide (GE Healthcare, USA) with multiple localized emitters was imaged. We split each channel and locate each emitter with ThunderSTORM (Fiji). Then a custom-written Matlab code used these coordinates to register the channels with two-dimensional affine transformation.

**Accuracy analysis.** The mapping error is calculated by $\frac{|V_O - V_A|}{V_R}$, in which $V_O$ is the observed value, $V_A$ is the accepted ground truth, and $V_R$ is the measurement range (1.2 for emission ratio, 1 for modulation depth, and π for dipole orientation). For simulation results in Fig. 2f, two fluorophores with same intensity are simulated with a distance of 300 nm in z-axis and with a small shift (50 nm) in x-axis. Different colors indicate the different optical properties of the fluorophores. The red fluorophore has an emission ratio of 1 and is oriented at 0° as an ideal dipole, and the yellow fluorophore has an emission ratio of 0.5 and is oriented at 60° as an ideal dipole. The simulated resolution for WF is 230 nm laterally and 670 nm axially, and the spatial resolution for SR is doubled for both spectrum and polarization measurement. The influence of noise on the measurement accuracies is further illustrated in Supplementary Fig. 8. For experimental results in Supplementary Figs. 3 and 4, we firstly label U2-OS cells only with Phalloidin-AF568, measure the of the AF568 fluorophores in areas free of out-of-focus signals, and taken the results as ground-truth values, since the optical properties of AF568 are stable in cellular environments. Then we label U2-OS cells with both Nile Red and Phalloidin-AF568, measure the optical properties under the influence of Nile Red signal, and calculate the mapping errors with different imaging modalities.

**Image analysis and statistical analysis.** The typical data processing flow of SPOT is illustrated in Supplementary Fig. 1 and Supplementary Note. Briefly, each detection channel was split and registered according to the alignment slide. Then the illumination nonuniformity is compensated for all six images. Afterward, every two images with the same pattern direction were processed with the HiLo algorithm[15], which produces three optical sectioned images under polarization modulation. The HiLo weight factor is set to $\alpha = 1$ for all experiments, and the influence of this factor is further illustrated in Supplementary Fig. 9. The SPOT intensity image for each channel was obtained by averaging the three optical sectioned images. The emission ratio was calculated by dividing the red channel image by the green channel image. The modulation depth and dipole orientation were calculated from the three polarization modulation images in the green channel by calculating the ±1st harmonics on the polarization dimension[13]. When processing the 3D-SIM data, the images of intensity and emission ratio were obtained from SIM reconstruction and the images of modulation depth and dipole orientation were obtained with SPOT algorithm. For each pattern direction, the uniform-illuminated image was obtained by averaging the images of five different phases, and the structure-illuminated image used the image with the first phase. The reconstruction was performed with custom-written codes in Matlab. The detailed data processing method is also included in this protocol[46], and the reconstruction code has been uploaded to Github.

When measuring the emission ratio and modulation depth of specific organelles, polylines were drawn manually on the intensity image and measured on the images of emission ratio and modulation depth in Fiji. To avoid the influence of motion drift on the measurement of modulation depth, the SPOT dataset was acquired repetitively and measured three times. The results would be accepted only when the measured fluctuation is within 15%. The type of compartments indicated by arrows throughout the paper was comprehensively determined by both their morphology and lipid properties. The plasma membrane, nuclear membrane, ER, and mitochondria with cristae structures were identified by their morphology. Other round-shaped compartments were determined when their polarity and phase were exclusively within one solid-line ellipse in Fig. 3e.

All statistical analyses were performed in GraphPad Prism and SPSS. The difference analysis between the two groups using unpaired, two-tailed $t$-test ($p < 0.05$) in Prism. Analysis of variance among multiple groups uses one-way ANOVA ($p < 0.05$) in SPSS.

**Reporting summary**. Further information on research design is available in the Nature Research Reporting Summary linked to this article.

## Data availability

The raw image files used in this paper are available via figshare with digital object identifier https://doi.org/10.6084/m9.figshare.13100471.v1[47]. Source Data are provided as a Source Data file. All other data that support the findings of this study are available from the corresponding author upon reasonable request.

## Code availability

The custom-written Matlab code[48] for SPOT is available on https://github.com/minor-planet/SPOT[48].

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

## Acknowledgements

This work was supported by National Natural Science Foundation of China (31971376, 61729501, 61831014, 61531014 and 61327902), National Key Research and Development Program of China (2017YFC0110202), the Beijing Natural Science Foundation (18JQ019), and Shenzhen Science and Technology Program (KQTD20170810110913065, ZDYBH201900000002, and JCYJ20180508152042002). The authors thank National Center for Protein Sciences at Peking University in Beijing, China, for assistance with SIM imaging.

## Author contributions

K.Z. conceived the project. P.X. and D.J. supervised the research. M.L., W.L., and K.Z. built the SPOT system. W.L., K.Z., X.C., and H.W. programmed the reconstruction software. M.L., K.Z., and C.S. performed Nile Red labeling experiments on homebuilt SPOT system and commercial system. Z.W., W.L., M.L., and K.Z. analyzed the data and do the statistical analysis. X.W. and X.C. prepared cardiac muscle cells with labeled Phalloidin-AF568. D.J., P.X., K.Z., M.L., W.L., and Q.D. wrote the manuscript with input from all the authors.

## Competing interests

The authors declare no competing interests.
