## [Peer Review File · Nature Communications]

Reviewers' Comments:

Reviewer #1:

Remarks to the Author:

This work by Zhanghao et al. aims at combining different microscopy modalities to image subcellular properties of lipid phases in a large variety of regions of live cells, including during cell division. The idea is to use a fluorescent lipid probe to report complementary information : polarity and phase (from polarization-resolved images). The work claims a gain in quantification of these properties by the use of super resolution, using a method based on polarized SIM previously published by the authors, combined with background rejection using HILO.

1- The idea to combine polarity with polarization methods is not new (contrary to what the authors seem to claim) and a lot of work has been done to set the bases of these approaches, and eventually understand the correlation between lipids orientational disorder and lipid phases defined as of different polarity in membranes : see for instance Cell Chemical Biology 9(11):1199-208 (2002) DOI: 10.1016/S1074-5521(02)00244-2, Biophys J. 78(1), P290-305 (2000) DOI : 10.1016/S0006-3495(00)76592-1, or the more general book Fluorescent Methods to Study Biological Membranes Eds. Yves Mely, Guy Duportail, Springer)

2- The methods are not presented in a clear way at the beginning of the manuscript : their specific relation to physical properties should be more clearly introduced (for instance why phase is directly related to polarization). Also the basis and principle of SPOT is not well described in the main text.

3- Polarization Optical Tomography (SPOT) is said to super resolve the membrane morphology, however it is misleading since the membrane morphology scale of interest (here < 100nm) for this topic is not clearly resolved by the method.

4- The authors want to emphasize that super resolution images are more appropriate for the quantification of membrane properties using polarity and phase estimations. This is indeed expected, however the shown comparisons are not appropriate : the authors use wide field images to base their argumentation, however most commonly sectioning methods are used that give much better contrasts and thus much better evaluation of the polarity/orientation properties in cell samples and even tissues. Since sectioning imaging is rather used, such as confocal or spinning disk microscopy, this should be the method to compare to SIM and SPOT.

5- Signal issues are not accounted for in the comparisons. Here accuracy arguments are developed on the basis of simulation supposing similar signals from different situation, however this is far from real since it is known that SIM deals with a lower level of signal. All the ingredients (signal, background etc) need to be accounted for in the comparison.

6- Since accuracy is at the centre of the work topic, measurements should be compared to true reference measurements. Such references exist in the literature, from known polarity-sensitive dyes and some known actin- phalloidin organized dyes. They should be used at least to convince the reader that the obtained values are close to the expected ones. Even Nile red is well known so the display of data in prepared known lipid phase samples should have been reported. This is an important step since it is known that polarity measurements can strongly depend on the method used and can be strongly biased from normalization procedures in noisy images, for instance.

7- Some open questions remain on the observations, for instance on the possible correlations between polarity and phase.

- the authors should specify which membrane leaflet they are probing with Nile Red since it is known that the membrane is not symmetric in polarity (and phase) behavior.

- the scale at which the observations are made could also be a factor that is considered in the conclusions, since phase changes are expected to occur at much lower scale than 50nm

- previous literature has pointed that the orientation of the dye could affect itself the emission spectrum of this molecule, therefore there could also be an influential effect on both properties, that cannot be considered as purely independent (see for instance Molecules2018,23, 1707; doi:10.3390/molecules23071707).

8- The polarization measurements reliability is not entirely clear : it is known from polarized fluorescence measurements that at least 4 polarization angles are required to avoid ambiguities in the resulted modulation and orientation quantities : why are only 3 angles sufficient (as read

from suppl fig S1)

- 9- SPOT and SIM3D should be compared in terms of longitudinal resolution : what are the numbers of this resolution ? this is missing in Fig. S2.
- 10- Figure 1 is poorly described in the text and its purpose is unclear. Some images of Fig. 1 for what concerns the GFP panels, are hardly visible.
- 11- Figure 2b,c : it is not clear if these figures are schematics or measurements. If they are measurements this needs to be specified together with the measurements conditions.
- 12- Figure 2f,g : simulations are performed to emphasize the improvement of accuracy for super resolution measurements versus wide field measurements. This graph is interesting, however comes from an improbable situation (considering the biological examples studied in this work). It would be more convincing to show an accuracy improvement on a known and calibrated sample, where signals measured would also integrate the environment of the cell (in particular with its background), the signal level (which is lower in super resolution imaging). Also The methods state that measurements are accepted if their time stability is better than 15% : therefore accuracies better than this level are not relevant, this experimental limit should be introduced in Fig. 2g.
- 13- The SPOT images exhibit a systematic dark contour around the features imaged (Fig S2) that seem like an artefact : this needs to be explained.
- 14- Fig. S3 : the authors compare their measurement to a ground-truth that is not clearly characterized : what are the ground truth values for polarity, phase and orientation ?
- 15- Fig. S3 : in this figure, why is the accuracy on polarity less affected by the sectioning capabilities ?
- 16- Fig. S4 should be compared with confocal and not WF. WF measurements are never performed to study sarcomeric structures in cardiac muscle cells.
- 17- It is not clear how the HILO processing affects signals and thus quantification of quantities such as signal modulation and color ratiometric measurements. The whole signal processing operation should be detailed in the supplementary materials, to show that processing does not harm the signal nature for quantification purposes such as done here.
- 18- The toxicity of Nile Red is not discussed, no is its internalization.
- 19- What is the effect of birefringence when performing polarization measurements in highly ordered samples ? this should affect accuracy as well and should be quite strong in, for instance, sarcomeric structures.

Reviewer #2:

Remarks to the Author:

The authors provide a smart approach for combining the Nile Red emission wavelength sensitivity on the polarity of membranes with SIM super-resolution and polarization fluorescence microscopy. By combining the two parameters polarization modulation depth and spectral emission changes, they can distinguish many different membrane types with a single dye. That method is smart and should be published.

I recommend that already in the abstract they better clarify how exactly they distinguish the ten different types of membranes, namely through the combination of these two parameters (Figure 3 e). Currently, the reader does not get from the statement in the abstract " This enables the high-throughput study of lipid heterogeneities of ten subcellular compartments, at different developmental stages, and even within the same organelle." what and how exactly the authors differentiate.

The study is also technically convincing. When reading the manuscript, it becomes convincingly clear how the authors can differentiate the different organelles based on the combination of Nile Red polarization modulation depth and spectral emission changes.

However, although the authors well acknowledge the literature on SIM and Nile Red, literature on fluorescence polarization microscopy is almost not cited, although their method depends largely on

fluorescence polarization microscopy. They should better acknowledge which groundwork previously existed and which are the new parts of their approach.

Finally, the authors also provide interesting examples of how their signals change during cell division. However, the extent to which new biophysical / biological insights are provided through this must be evaluated by reviewers who are experts in cell division.

Minor comments:

Page 4 line 74: What do the authors mean by enormous in " Membranes in mammalian cells consist of enormous types of lipids"?

Page 10 line 205: "In this paper, we manually identify plasma membrane, nuclear membrane, TNT, ER, and mitochondria".... What do the authors mean by manually?

In summary, I recommend publishing when it is clear which new biophysical / biological insight about cell division are gained and after previous work about SIM, Nile red and fluorescence polarization microscopy is recognized in a more balanced way.

Reviewer #3:

Remarks to the Author:

In this manuscript, Zhanghao et al. obtained fine image of emission and polarization modulation of Nile red using SPOT technique. The data are interesting. However, the authors have oversimplified view of physical properties of lipids. Their results clearly show that there is no correlation between cholesterol/sphingolipid content and emission ratio/modulation depth. Extensive lipidomics has to be accompanied to reach "a new horizon in correlating single-cell super-resolution lipidomics with multiplexed imaging of organelle interactome".

Fig. 3.

- Distribution of each organelle marker has to be shown.
- In this resolution, it is not clear whether the obtained difference in mitochondria is due to the cristae and other membranes or the heterogeneous lipid distribution in mitochondria.
- Late endosomes are multivesicular organelle with different lipid composition. Does their method distinguish heterogeneity of the organelle?
- Why emission ratio is higher but modulation depth is lower in LE?
- Does their method detect heterogeneity of plasma membrane?
- What are alphabets in Fig. 3c and d?
- "During maturation, late endosomes lose cholesterol and sphingolipid" Does this mean degradation of sphingolipid in late endosomes and cholesterol exit by NPC1? Phospholipids are also degraded in late endosomes/lysosomes.

Fig. 4.

- What are alphabets in Fig. 4c?
- Detailed explanation of the methods is required. How reproducible are these results?

Response to Reviewers' Comments

We very much appreciate the critical reading of our manuscript and valuable suggestions by the reviewers. We have carefully reviewed the comments and have revised the manuscript accordingly. The revised text is underscored and is also included here. The responses to the comments are listed point-by-point as follows in blue (the page and line numbers refer to the revised manuscript):

Reviewer #1

This work by Zhanghao et al. aims at combining different microscopy modalities to image subcellular properties of lipid phases in a large variety of regions of live cells, including during cell division. The idea is to use a fluorescent lipid probe to report complementary information: polarity and phase (from polarization-resolved images). The work claims a gain in quantification of these properties by the use of super resolution, using a method based on polarized SIM previously published by the authors, combined with background rejection using HILO.

Response: We thank the reviewer for very detailed assessment of our work that significantly improves the quality of our manuscript. Below are the responses to specific issues one by one.

Q1

The idea to combine polarity with polarization methods is not new (contrary to what the authors seem to claim) and a lot of work has been done to set the bases of these approaches, and eventually understand the correlation between lipids orientational disorder and lipid phases defined as of different polarity in membranes: see for instance Cell Chemical Biology 9(11):1199-208 (2002) DOI: 10.1016/S1074-5521(02)00244-2, Biophys J. 78(1), P290-305 (2000) DOI : 10.1016/S0006-3495(00)76592-1, or the more general book Fluorescent Methods to Study Biological Membranes Eds. Yves Mely, Guy Duportail, Springer)

Response: We thank the reviewer for bringing these publications to our attention, and we agree with the reviewer that researchers had paid much attention to study the correlation between the lipid phase and polarity. In the reference of *Biophys J. (2000)*, the authors performed spectral

measurements (the excitation generalized polarization, GP) on different lipid mixtures at various temperatures. Because the two-photon excitation used in their experiments was linearly polarized, they also observed that high polarization-selected excitation was associated with high GP (or low polarity). In the reference of *Cell Chemical Biology (2002)*, the authors synthesized several two-band ratiometric fluorescence probes, which located and orientated differently in membranes. They measured the fluorescence spectra of the probes and studied the relation between fluorescence spectra and the probe location.

In these experiments, nevertheless, only the fluorescence spectrum was measured. Some other publications only measured the fluorescence polarization, such as Ref. 8-11 in the manuscript. **Our work is the first to obtain simultaneous spectrum and polarization measurements, and to extend these measurements with organelle resolution.** In the reference of *Biophys J. (2000)* and many other publications, observations seemed to imply that a low polarity always resulted in a lipid-order phase. However, this conclusion was mostly based on experiments with giant unilamellar vesicles composed of one or two types of lipids. In mammalian cells, there exist at least 10,000 different lipids in three categories (glycerophospholipid, sphingolipid, and cholesterol), and the lipid phase is strongly related to the lipid-lipid interaction. In our results, the lipid phase increases as the lipid polarity drops in mitochondria, late endosome, lysosome, early endosome, and plasma membrane. However, counterexamples also exist. For example, lipid droplets have both low lipid polarity and phase because the core stores the low-polarity lipid, such as triglyceride in disorder. Therefore, the simultaneously measured emission ratio and modulation depth by SPOT can directly indicate the lipid polarity and phase and serve as complementary information.

To avoid confusion and to provide a more detailed discussion, we revised the related texts as follows:

Revised texts in Line 18, Page 12:

Advantages of SPOT. Previous studies only measured either the lipid polarity^{19,23,24} or the lipid phase⁸⁻¹¹. Some of these studies assumed that a higher lipid polarity led to a lower degree in lipid order or the other way around. This assumption is correct in the synthesized vesicles with one or two types of lipids. However, lipid diversity is much more complex in mammalian cells that produce more than 10,000 different types of lipids. Besides, the lipid phase is strongly associated with the lipid-lipid interactions², so that it should not be solely determined by the lipid polarity. Using SPOT, we presented for the first time the capability of correlative imaging of both the emission ratio and modulation depth, to quantify the complementary information of lipid polarity

and phase. In our results, as the lipid polarity drops, the lipid phase increases in mitochondria, late endosome, lysosome, early endosome, and plasma membrane, which is consistent with the above assumption. The lower emission ratio suggests a higher proportion of sphingolipid or cholesterol that results in a higher lipid order. However, lipid droplet serves as a counterexample that shows both low lipid polarity and order. This is because the core of lipid droplets stores the low-polarity lipid, such as triglyceride in disorder. Nuclear membrane and ER have high lipid polarity, but with a relatively high ordered phase, which may be related to the sophisticated lipid-lipid interactions. The wide existence of lipid heterogeneity uncovered by SPOT demonstrates its complementary capacity to mass spectroscopy^{25,26} in quantifying the molecular composition of lipids, as mass spectroscopy is insufficient for live-cell studies.

Q2

The methods are not presented in a clear way at the beginning of the manuscript: their specific relation to physical properties should be more clearly introduced (for instance why phase is directly related to polarization). Also the basis and principle of SPOT is not well described in the main text.

Response: We thank the reviewer for these comments. We have added more details about how the spectral and polarization properties are related to the lipid phase and polarity. And we also have the basic principle of SPOT described in the main text, which is further illustrated in the Supplementary Note and Fig. S1.

Revised description of the relationship between the optical properties and lipid properties in Line 3, Page 4:

Two distinct physical properties of Nile Red are first explored in our work: the emission spectrum changes are measured by two-color ratiometric imaging (Fig. 2b), and the wobbling dynamics of fluorescent dipoles are resolved by polarization modulation (Fig. 2c). The emission spectrum of Nile Red will red shift in polar environment and blue shift in nonpolar environment⁶. Membranes in mammalian cells consist of three lipid categories: glycerophospholipid, sphingolipid, and cholesterol (Fig. 2d, e)². The lipid polarity descends from unsaturated glycerophospholipid, saturated glycerophospholipid, sphingolipid, to cholesterol, so that the emission spectrum of Nile Red will shift from red to blue accordingly. The sphingolipid-rich membrane tends to form the solid-like phase, the membrane rich of cholesterol and saturated glycerophospholipid tends to form the liquid-ordered (Lo) phase, and the membrane rich of unsaturated glycerophospholipid tends to form the liquid-disordered (Ld) phase^{1,7}. Polarization imaging of fluorophores can reveal the orientational wobbling of molecular dipoles to distinguish ordered or disordered lipid phases⁸⁻¹¹. In the ordered membrane, the wobbling of dipoles is more restricted so that Nile Red exhibits stronger polarization response with increased modulation depth. In disordered membrane, the wobbling of dipoles is more flexible so that Nile Red exhibits weaker polarization response with decreased polarization modulation depth.

Revised illustration of SPOT principle in the main text, Line 3, Page 5:

Therefore, we have further developed a Spectrum and Polarization Optical Tomography (SPOT) method (details in Supplementary Fig. 1, Supplementary Note) to increase the measuring accuracies for both lipid polarity and phase (Supplementary Fig. 2-4). The principle of SPOT is to obtain optical sectioning with structured illumination^{14,15} with two or more phase shifts. Using the HiLo method¹⁵ by fusing the High and Low spatial frequency information for each pattern direction, the less modulated out-of-focus background is rejected from the in-focus signals. Similar to polarized SIM¹³, three pattern directions are applied for polarization modulation analysis, so that SPOT requires six images with structured illumination, less than the typical nine images required by SIM. Besides, SPOT can also be applied to the SIM dataset and take advantage of the doubled spatial resolution.

Q3

Polarization Optical Tomography (SPOT) is said to super resolve the membrane morphology, however it is misleading since the membrane morphology scale of interest (here < 100nm) for this topic is not clearly resolved by the method.

Response: Yes. Limited by the resolution of SIM, SPOT can only obtain an averaged emission ratio and a decreased polarization modulation depth for features below 100 nm. We have added this point to the discussion session.

Added discussion in Line 9, Page 13:

Limitations of SPOT. Besides the optical sectioning in the axial dimension, the lateral resolution remains diffraction-limited for SPOT, while the resolution of SPOT-SIM3D is doubled. As a result, the polarity measurement is the averaged value of the lipid assembly within the point spread function (PSF). When curved membranous structures are within the PSF, the measurement will result in a significantly lower polarization modulation depth due to an ensemble dipole of the different molecular dipole orientations. For example, the membranous structure of Golgi apparatus is crowded and cannot be clearly resolved by SPOT, which further leads to a reduced lipid order in the measurement of Golgi apparatus.

Q4

The authors want to emphasis that super resolution images are more appropriate for the quantification of membrane properties using polarity and phase estimations. This is indeed expected, however the shown comparisons are not appropriate: the authors use wide field images to base their argumentation, however most commonly sectioning methods are used that give much better contrasts and thus much better evaluation of the polarity/orientation properties

in cell samples and even tissues. Since sectioning imaging is rather used, such as confocal or spinning disk microscopy, this should be the method to compare to SIM and SPOT.

Response: We thank the reviewer for pointing out other optical sectioning techniques such as confocal or spinning disk (SD). In our manuscript, we compare WF and SPOT for several reasons: (1) the WF and SPOT data can be acquired in the same system so that these results can be compared under the same condition. (2) Although spectral ratiometric imaging with confocal/SD confocal is easy to perform, polarization imaging with (SD) confocal requires specialized instruments for instance mentioned in Ref. 8-11. (3) The imaging speed of confocal is not as fast as SIM, which will become an issue, especially for polarization imaging, because motion will bias the polarization results as discussed in the manuscript.

Based on the above discussion that the imaging speed of confocal fails to catch the polarization imaging in live cells. **We compare the spectral measurements between confocal and SPOT/SPOT-SIM3D in fixed cells.** We label U2OS cells with both NileRed and actin-phalloidin AF568, and the optical properties of AF568 are measured and serve as ground truth. The polarity measurement accuracy of confocal imaging is 91% and is added in Supplementary Fig. 3c:

Fig. R1 – The images of emission ratio obtained with a commercial confocal microscope (Leica TCS SP8, pinhole: 1 a.u.). U2-OS cells in the left image are stained with both Nile Red and Phalloidin-AF568, and the cells in the right image are only stained with Phalloidin-AF568. The

ground truth of the AF568 emission ratio is measured on the lines of the right image. The accuracy measurement is performed on the lines of the left image.

Revised Supplementary Fig. 3:

Supplementary Fig. 3 – Comparison of the measurement accuracies in the emission ratio and modulation depth among WF, Confocal, SIM3D, SPOT, and SPOT-SIM3D. (a) The WF image of U2-OS cells with actin filaments labeled by Phalloidin-AF568. We measured the emission ratio, modulation depth, and dipole orientation of the AF568 fluorophores in areas free of out-of-focus signals (marked by the orange lines) and taken them as ground-truth values. (b) The WF image of U2-OS cells with the actin filaments labeled by Phalloidin-AF568 and the lipid membranes stained by Nile Red. The measurement of the optical properties of the AF568 fluorophores is strongly disturbed by the out-of-focus signals of Nile Red that stains across the crowded subcellular compartments. (c-e) The histograms compare the emission ratio (c), modulation depth (d), and dipole orientation (e) among WF, SIM3D, SPOT, and SPOT-SIM3D. SIM3D, SPOT, and SPOT-SIM3D achieved significantly higher accuracies than WF when measuring the emission ratio of AF568 fluorophores in Nile Red stained cells. SPOT and SPOT-SIM3D achieved significantly higher measurement accuracies in the modulation depth. Because the out-of-focus signals generated by the dipoles of Nile Red are polarization isotropic, all the imaging methods achieve high measurement accuracy in the dipole orientation. To further compare SPOT with confocal microscopy, we measured the emission ratio on a commercial confocal microscope (Leica TCS SP8, pinhole: 1 a.u.). The measurement accuracy of confocal in emission ratio (c) is also improved compared with WF but slightly lower than SPOT. However,

we cannot conclude that SPOT is better than confocal because the measurements of emission ratios were performed on two microscopes that may introduce other influencing factors. Also, the polarization measurement accuracy of confocal is not included here because polarization imaging is not available on commercial confocal microscopes. Scale bar: 5 μm .

Q5

Signal issues are not accounted for in the comparisons. Here accuracy arguments are developed on the basis of simulation supposing similar signals from different situation, however this is far from real since it is known that SIM deals with a lower level of signal. All the ingredients (signal, background etc.) need to be accounted for in the comparison.

Response: We thank the reviewer for the careful assessment of our work. Both the signal level of fluorescence and the noise level of the detector will influence the measurements. The noise model of the sCMOS camera has been well characterized elsewhere (*F. Huang, et al., Nature methods 10 (7), 653-658 (2013)*). In our experiments, the baseline for each pixel of the sCMOS camera was obtained by averaging a series of images acquired with no excitation and was subtracted before polarization and spectrum analysis. Although other noise sources such as shot noise, thermal noise, readout noise, etc. remains throughout the reconstruction, signal-to-noise ratio (SNR) is the key factor affecting the measurement. Empirically we require the raw image of SIM with a peak signal-to-noise ratio (PSNR) of 20 dB, which leads to much higher PSNR in reconstructed images. Therefore, we added a simulation to compare the accuracy of SIM measurements with PSNR from 10 dB to 30 dB with wide-field result without noise.

In the simulation experiment, we added different Gaussian noises to form SIM images with different PSNR. The simulation parameters are set as shown in Methods. Supplementary Fig. 8 shows the influence of noise on measurement results. The measurement accuracies of noise-free WF images are 0.85, 0.55 and 0.81 for polarity, phase and orientation. The measurement accuracies of SIM are influenced by the noise level as shown in Supplementary Fig. 8d-f. We can see that the measurement accuracy increases with the increase of PSNR, while the decrease of std indicates the improvement of measurement robustness. When the PSNR is greater than 17 dB, all the accuracies of SIM are better than that of wide-field, which are 0.87, 0.88, 0.97 for polarity, phase and orientation measurement.

Added Supplementary Fig. 8:

Supplementary Fig. 8 – Influence of noise on measurement accuracies. (a, b) The WF and SR images under the excitation polarizations of 0°, 60°, and 120°. (c) The images correspond to (b) with a PSNR of 20 dB. For each PSNR level, we repeat 500 trials of simulation and study the influence of noise on measurement accuracies with two statistical indicators: the mean accuracy and the standard deviation (std) value of the accuracy. The std of accuracy means how robust under noise the measurements are, which can also be termed as ‘precision’. (d) The relationships between polarity accuracy and PSNR in SR measurements. The blue line represents the relationship between the mean accuracy and PSNR, while the orange line represents the relationship between the precision and PSNR. (e, f) The changes of phase accuracy and orientation accuracy with PSNR, respectively. The measurement accuracies of noise-free WF images are 85%, 55%, and 81% for polarity, phase, and orientation. When the PSNR is greater than 17 dB, all the accuracies of SR measurements are better than those of wide-field, which are 87%, 88%, and 97%. In this work, we empirically require a PSNR of >20 dB for the raw image acquired throughout all imaging experiments.

Q6

Since accuracy is at the centre of the work topic, measurements should be compared to true reference measurements. Such references exist in the littérature, from known polarity-sensitive dyes and some known actin-phalloidin organized dyes. They should be used at least to convince the reader that the obtained values are close to the expected ones. Even Nile red is well known so the display of data in prepared known lipid phase samples should have been reported. This is an important step since it is known that polarity measurements can strongly depend on the

method used and can be strongly biased from normalization procedures in noisy images, for instance.

Response: We thank the reviewer for these suggestions. Firstly, we further describe the detailed SPOT reconstruction process in the Supplementary Note. The whole reconstruction only contains one adjustable parameter, the HiLo weighting factor α , and the parameter has little influence on the measurement accuracies (demonstrated in added Fig. S9). Besides, we choose $\alpha=1$ throughout all experiments.

Secondly, we performed calibration experiments with polarity-known solvents, which is helpful to relate measurement results on different microscopes. We calibrated the relationship between the emission ratio of Nile Red and the polarity in several solvents with known polarity (see: *Chemical Reviews* 94, 2319-2358 (1994) DOI:10.1021/cr00032a005), as shown in added Supplementary Fig. S7. The polarity of the solvent is indicated by the molar transition energy ($E_T(30)$). We can see that the polarity of the solution increases as the emission ratio increase. The results are consistent with the conclusions on emission spectrum measurement in the reference: *J Phys Chem Lett* 10, 2414-2421 (2019) DOI: 10.1021/acs.jpcllett.9b00668. We further fitted the relationship between emission ratio and solvent polarity with linear regression. Then we calculated the lipid droplet polarity through the emission ratio we measured. The result is in good agreement with that in the reference: *Analytical Chemistry* 91, 1928-1935 (2019)DOI: 10.1021/acs.analchem.8b04218.

Added Supplementary Fig. 7:

Supplementary Fig. 7 – Calibration of the Nile Red emission ratio in solvents with known polarity. We dissolve Nile Red in solvents, including ethyl acetate, acetone, DMF, DMSO, ethanol, and methanol, whose polarities of solvents are known and indicated by the molar transition energy $E_T(30)$. The emission ratios in solvents are measured and fitted as in (a, b). (b) The average polarities of different compartmental membranes are further calculated based on the fitted curve and the measured emission ratio. The $E_T(30)$ polarity of the lipid droplet is 38.335 that is similar to that in the previous publication¹.

Added Supplementary Fig. 9:

Supplementary Fig. 9 – Influence of the HiLo weighting factor on measurement results. (a) The WF image. (b-d) The HiLo images with the HiLo weighting factors (α) of 1.0, 2.0, and 3.0, respectively. (e-f) The relationship of polarity, phase, and orientation accuracy with the HiLo weighing factor α . The measurement accuracies in the emission ratio, polarization modulation depth, and dipole orientation are little changed when the HiLo weighting factors (α) varies from 0.5 to 3. We set the HiLo weighting factors (α) to 1 throughout all imaging experiments.

Q7

Some open questions remain on the observations, for instance on the possible correlations between polarity and phase.

Response: Thank the reviewer for the valuable suggestion. The lipid polarity and phase are both determined by the composition of lipids and their biochemical properties. In GUVs composed of one or two types of lipids, a lower lipid polarity often leads to a higher lipid order (and vice versa). However, the lipid composition and lipid-lipid interactions are much more complex, and the lipid phase is not solely related to the lipid polarity. With the simultaneous measurement of the lipid polarity and phase by SPOT, we find that the nuclear membrane and ER have both high lipid polarity and phase, and lipid droplets have both low lipid polarity and phase. We added some discussion in our manuscript as follows:

Added discussion in Line 18, Page 12:

Advantages of SPOT. Previous studies only measured either the lipid polarity^{19,23,24} or the lipid phase⁸⁻¹¹. Some of these studies assumed that a higher lipid polarity led to a lower degree in lipid

order or the other way around. This assumption is correct in the synthesized vesicles with one or two types of lipids. However, lipid diversity is much more complex in mammalian cells that produce more than 10,000 different types of lipids. Besides, the lipid phase is strongly associated with the lipid-lipid interactions², so that it should not be solely determined by the lipid polarity. Using SPOT, we presented for the first time the capability of correlative imaging of both the emission ratio and modulation depth, to quantify the complementary information of lipid polarity and phase. In our results, as the lipid polarity drops, the lipid phase increases in mitochondria, late endosome, lysosome, early endosome, and plasma membrane, which is consistent with the above assumption. The lower emission ratio suggests a higher proportion of sphingolipid or cholesterol that results in a higher lipid order. However, lipid droplet serves as a counterexample that shows both low lipid polarity and order. This is because the core of lipid droplets stores the low-polarity lipid, such as triglyceride in disorder. Nuclear membrane and ER have high lipid polarity, but with a relatively high ordered phase, which may be related to the sophisticated lipid-lipid interactions.

- the authors should specify which membrane leaflet they are probing with Nile Red since it is known that the membrane is not symmetric in polarity (and phase) behavior.

Response: Nile Red labels both the inner and outer leaflets of the lipid bilayer according to previous publications (See: *Journal of the American Chemical Society* 132, 4907-4916, (2010) DOI: 10.1021/ja100351w). Therefore, we measured the average lipid properties of both leaflets.

- the scale at which the observations are made could also be a factor that is considered in the conclusions, since phase changes are expected to occur at much lower scale than 50nm

Response: Yes, SPOT cannot reveal lipid heterogeneity below 100 nm. SPOT is suitable to observe the lipid heterogeneity at the scale larger than 100 nm, but luckily, the size of most organelles is larger than 100 nm. We have added a related discussion in the revised manuscript.

Added discussion in Line 9, Page 13:

Limitations of SPOT. Besides the optical sectioning in the axial dimension, the lateral resolution remains diffraction-limited for SPOT, while the resolution of SPOT-SIM3D is doubled. As a result, the polarity measurement is the averaged value of the lipid assembly within the point spread function (PSF). When curved membranous structures are within the PSF, the

measurement will result in a significantly lower polarization modulation depth due to an ensemble dipole of the different molecular dipole orientations. For example, the membranous structure of Golgi apparatus is crowded and cannot be clearly resolved by SPOT, which further leads to a reduced lipid order in the measurement of Golgi apparatus.

- previous literature has pointed that the orientation of the dye could affect itself the emission spectrum of this molecule, therefore there could also be an influential effect on both properties, that cannot be considered as purely independent (see for instance Molecules2018,23, 1707; doi:10.3390/molecules23071707).

Response: In our work, the optical properties of different lipid membranes are simultaneously measured with the same dye Nile Red and under the same condition, which minimizes the influencing factors to reach convincing conclusions. The valuable reference mentioned by the review drew the conclusion based on a theoretical simulation that the dipole orientation may influence its emission spectrum. We believe the characterization of the dyes in in-vitro lipid vesicles with known lipid composition can provide a quantitative assessment on this question. However, the focus of our work is to demonstrate the increased accuracies of SPOT in measuring the spectrum and polarization signals and to reveal universal lipid heterogeneity among subcellular compartments. Studying the relationship between the optical properties of Nile Red and the lipid characteristics will remain to be our future work.

Added discussion in Line 16, Page 13:

Another limitation lies in the correlation between the measured optical properties and the lipid properties of the membrane, which is related to the dye used. Though we have carefully calibrated the emission ratio of Nile Red in different solvents with known polarity (Supplementary Fig. 7), the environment of cellular lipid membranes is much more sophisticated. Therefore, the penetration depth and the orientation of the dye may differ in the membranes of different lipid compositions, which could influence its emission ratios³⁴. Fortunately, the optical properties of different lipid membranes are simultaneously measured with the same dye Nile Red and under the same condition, which minimizes these influencing factors. Future work better includes characterizations of the dyes in in-vitro lipid vesicles with known lipid composition, which can provide more insights on this discussion.

SPOT and SIM3D should be compared in terms of longitudinal resolution: what are the numbers of this resolution? this is missing in Fig. S2.

Response: We thank the reviewer for the constructive suggestion. We have added the comparison of axial resolution between WF, Confocal, SPOT, and SPOT-SIM3D in Supplementary Fig. 2b. The longitudinal resolution of SPOT is comparative to confocal, and SIM3D achieves a doubled axial resolution.

Revised Supplementary Fig. 2:

Supplementary Fig. 2 – The improved optical sectioning of SPOT and its enhanced measurement accuracies in both emission ratio and polarization modulation depth. (a) The

comparison among wide-field (WF), confocal, SPOT, and SIM on the XZ images of 100-nm fluorescent beads. The SPOT result shows a significant attenuation of the out-of-focus signal that is comparable to confocal result acquired in a commercial system (Leica TCS SP8, pinhole: 1 a.u.). **(b)** The normalized intensity profiles along the z-axis in **(a)** quantify the power of optical sectioning, and the z intensity profile of a single bead quantifies the longitudinal resolution of different imaging modalities. **(c-d)** the comparison results of the lipid polarity map and the lipid phase map of mitochondria between WF (left) and SPOT (right) modalities. **(e-f)** the comparison results of the lipid polarity map and the lipid phase map of lipid droplets. Scale bar: (a) 5 μm ; (c-d) 5 μm ; (e-f) 2 μm .

Q9

Figure 1 is poorly described in the text and its purpose is unclear. Some images of Fig. 1 for what concerns the GFP panels, are hardly visible.

Response: Thank you, and we agree with your comments. Fig. 1 demonstrates that Nile Red can sufficiently stain the lipid membranes on various subcellular compartments. Besides, Fig. 1 also shows how we determine the specific type of compartments. We have enriched the description of Fig. 1 in our manuscript. Because Fig. 1 contains colocalization images of many compartments, we also included a high-res version of Fig. 1 in supplementary materials for better visualization.

Revised texts in Line 20, Page 2:

Here we employ Nile Red⁵, a common intracellular lipid dye, to stain lipid membranes universally in live cells (Fig. 1). From the GFP colocalization images and the Nile Red membrane morphology images, we identify that Nile Red successfully labels at least ten subcellular compartments. Fig. 1b shows that the mitochondria-GFP, Golgi-GFP, ER-GFP, lysosome-GFP, early endosome-GFP, and late endosome-GFP colocalize the corresponding organelles. Since Nile Red emits green-yellow fluorescence only in lipid droplets, colocalization of Nile Red signals in the green and yellow channels indicates the organelle type of lipid droplets (Fig. 1c). The plasma membrane, nuclear membrane, and tunneling nanotubule can be easily identified from their morphology in the intensity image of Nile Red (Fig. 1d-f).

Q10

Figure 2b,c: it is not clear if these figures are schematics or measurements. If they are measurements this needs to be specified together with the measurements conditions.

Response: These figures are schematic demonstration, and we have made it clear in the figure caption.

Q11

Figure 2f,g: simulations are performed to emphasize the improvement of accuracy for super resolution measurements versus wide field measurements. This graph is interesting, however comes from an unprobable situation (considering the biological examples studied in this work). It would be more convincing to show an accuracy improvement on a known and calibrated sample, where signals measured would also integrate the environment of the cell (in particular with its background), the signal level (which is lower in super resolution imaging). Also The methods state that measurements are accepted if their time stability is better than 15%: therefore accuracies better than this level are not relevant, this experimental limit should be introduced in Fig. 2g.

Response: Yes, the reviewer is right that accuracy measurements on cell specimen is required. We use the spectrum and polarization signal of Phalloidin-Alexa Flour 568 labeled on the actin filaments as a reference specimen. We choose this reference specimen for several reasons. Firstly, AF568 has a stable emission spectrum in the cellular environment. Secondly, the polarization of phalloidin conjugated AF-568 on actin filaments has been well characterized (*Proc Natl Acad Sci* 113, e820-e828 (2016) DOI: 10.1073/pnas.1516811113, *Light: Science & Applications* 5 (10), e16166 (2016) DOI: 10.1038/lssa.2016.166, etc.), which also exhibits stable modulation depth. Thirdly, we can easily find sparse and thick actin filament in WF images so that its spectrum and polarization can be measured without HiLo processing. We carried out the statistical measurements in the areas free of out-of-focus signals and got the mean values of polarity, phase, and orientation as the ground truth values. Afterward, we stain the cells with both Nile Red and Phalloidin-AF568 and compare the measured polarity, phase, and orientation with the ground-truth results.

The 15% stability of the polarization modulation depth is required to exclude biased measurements caused by the movement of cells. Fig. 2g is simulation results which do not have the problem of movement. The detailed description of how the time stability is checked is included in the Supplementary Note as follows.

Added text in Supplementary Note, section g) ii, Page 15:

Check the fluctuation of modulation depth. This step is mainly used to exclude the data that motion blur or photobleaching have a significant impact on the measurement. As shown in step f), at least 3 polarization modulations are required to get modulation depth *OUF* and dipole

orientation β , and we adopted $\theta_i = 0^\circ, 60^\circ, 120^\circ$ in the experiment. For the data check, we carry out 5 polarization modulations in sequence with $\theta_i = 0^\circ, 60^\circ, 120^\circ, 0^\circ, 60^\circ$. When selecting $\theta_i = 0^\circ, 60^\circ, 120^\circ$, we can find the OUF . Similarly, when choosing $\theta_i = 60^\circ, 120^\circ, 0^\circ$ and $\theta_i = 120^\circ, 0^\circ, 60^\circ$, we can also solve the modulation depth, expressed as OUF' and OUF'' . OUF , OUF' and OUF'' should be close to each other if the motion blur and photobleaching are not obvious. We excluded the data where OUF , OUF' and OUF'' differ more than 15%.

Q12

The SPOT images exhibit a systematic dark contour around the features imaged (Fig S2) that seem like an artefact: this needs to be explained.

Response: We believe that the reviewer refers to the images in Supplementary Fig 2e, f. We included the raw data of Supplementary Fig. 2 here without any post processing, which shows no obvious artifacts. The blurred out-of-focus signals were diminished in the HiLo images taking advantage of its optical sectioning capability.

Figure R2 – The raw data of Fig. S2. (a) WF image, (b) HiLo image, (c-d) the magnified view of (a) and (b), respectively.

Q13

Fig. S3: the authors compare their measurement to a ground-truth that is not clearly characterized: what are the ground truth values for polarity, phase and orientation?

Response: Thank the reviewer for pointing our this. We used the spectrum and polarization signal of Phalloidin-Alexa Flour 568 labeled on the actin filaments as a reference specimen. We chose this reference specimen for several reasons. Firstly, AF568 has a stable emission spectrum in cellular environment. Secondly, the polarization of phalloidin conjugated AF-568 on actin

filaments has been well characterized (*Proc Natl Acad Sci* 113, e820-e828 (2016) DOI: 10.1073/pnas.1516811113, *Light: Science & Applications* 5 (10), e16166 (2016) DOI: 10.1038/lisa.2016.166, etc.), which also exhibits stable modulation depth. Thirdly, we can easily find sparse and thick actin filament in WF images so that its spectrum and polarization can be measured without HiLo processing. We carried out the statistical measurements in the areas free of out-of-focus signals and got the mean values of polarity, phase and orientation as the ground truth values. We added a description of the ground truth values both in online Methods.

Revised texts in online Methods (Line 6, Page 20):

The influence of noise on the measurement accuracies is further illustrated in Supplementary Fig. 8. For experimental results in Supplementary Fig. 3, 4, we firstly label U2-OS cells only with Phalloidin-AF568, measure the of the AF568 fluorophores in areas free of out-of-focus signals, and taken the results as ground-truth values, since the optical properties of AF568 are stable in cellular environments. Then we label U2-OS cells with both Nile Red and Phalloidin-AF568, measure the optical properties under the influence of Nile Red signal, and calculate the mapping errors with different imaging modalities.

Q14

Fig. S3: in this figure, why is the accuracy on polarity less affected by the sectioning capabilities?

Response: In many cases, polarization measurements are more vulnerable to the out-of-focus signal than the spectrum ratio (polarity). For example, for a pixel with emission ratio 0.5 (two-channel intensities as $C1_{gt}=1500$, $C2_{gt}=3000$); with polarization modulation depth of 0.5 (non-modulated intensity $DC_{gt}=2000$, modulated intensity $AC_{gt}=1000$, so that the peak signal is 3000 and equals to $C2_{gt}$). When the out-of-focus signal is 1000, the measured values of the two detection channels and the AC value of the polarization response signal will increase, while the DC value of the polarization response signal will remain unchanged. Then the measured emission ratio would be $2500/4000=0.625$; the measured polarization modulation depth would be $1000/3000=0.333$, and their corresponding accuracies are 0.90 and 0.83, respectively. The comparison result of polarity and modulation depth measurement is shown in Fig. R3.

Fig. R3 – Comparison of the sensitivity of polarity and phase measurements to the same out-of-focus noise. (a) The optical-sectioning images obtained from two detection channels. (b) The optical-sectioning images under 3 different excitation polarizations (θ), the corresponding excitation polarization from left to right are 0° , 60° , 120° . (c), (d) The images corresponding to (a), (b) when there is no optical sectioning capability. (e) Polarization measurement results of the image center point. $C1_{gt}$ and $C2_{gt}$ represent the detection intensities of the two detection channels in the optical-sectioning images, while $C1_{bg}$ and $C2_{bg}$ are the corresponding detection intensities with out-of-focus signals. R_{bg} and R_{gt} are polarity measurements with and without out-of-focus signals, and $Acc1$ represents the polarity accuracy with the out-of-focus noise. (f) Phase measurement results of the image center point. AC_{gt} , DC_{gt} , and M_{gt} represent AC, DC, and phase measurement in the optical sectioning images, respectively, while AC_{bg} , DC_{bg} , and M_{bg} are corresponding values when there exists out-of-focus noise. $Acc2$ represents the phase accuracy with the out-of-focus noise.

Q15

Fig. S4 should be compared with confocal and not WF. WF measurements are never performed to study sarcomeric structures in cardiac muscle cells.

Response: We have added the comparison between confocal and SPOT/SPOT-SIM3D with U2-OS cells. Please also see our reply to Q4. However, we are sorry that we cannot provide the

confocal results of cardiac muscle cells. The specimen is provided by our collaborators in Beijing and is currently not available due to the COVID-19 pandemic.

Q16

It is not clear how the HILO processing affects signals and thus quantification of quantities such as signal modulation and color ratiometric measurements. The whole signal processing operation should be detailed in the supplementary materials, to show that processing does not harm the signal nature for quantification purposes such as done here.

Response: Thanks. We described the detailed SPOT reconstruction process in Supplementary Note. The whole reconstruction only contains one adjustable parameter, the HiLo weighting factor α . The parameter has little influence on the measurement accuracies (demonstrated in added Fig. S9). Besides, we choose $\alpha=1$ throughout all experiments.

Added Supplementary Fig. 9:

Supplementary Fig. 9 – Influence of the HiLo weighting factor on measurement results. (a) The WF image. (b-d) The HiLo images with the HiLo weighting factors (α) of 1.0, 2.0, and 3.0, respectively. (e-f) The relationship of polarity, phase, and orientation accuracy with the HiLo weighing factor α . The measurement accuracies in the emission ratio, polarization modulation depth, and dipole orientation are little changed when the HiLo weighting factors (α) varies from 0.5 to 3. We set the HiLo weighting factors (α) to 1 throughout all imaging experiments.

Q18

The toxicity of Nile Red is not discussed, no is its internalization.

Response: We thank the reviewer for pointing out this. Nile Red had been demonstrated as a biocompatible dye for the study of lipid dynamics in living cells in Ref 5, 12, and we have added the discussion of the bio-compatibility of Nile Red.

Added text in Line 18, Page 4:

Besides these multiple responsive optical properties, the fluorogenic dye Nile Red is highly bio-compatible^{5,12} and only emits fluorescence within the hydrophobic environment of lipid membranes, which is highly suitable for long-term monitoring of lipid dynamics in living cells.

Q19

What is the effect of birefringence when performing polarization measurements in highly ordered samples? this should effect accuracy as well and should be quite strong in, for instance, sarcomeric structures.

Response: We thank the reviewer for the discussion. Birefringence does exist in tissue specimens if there are highly ordered structures, which will degrade the fluorescence anisotropy measurement. The specimen of live U2-OS cells has no birefringence effect as no such ordered structure can be formed. Even the specimen of single cardiac muscle cell presents little birefringence because the measured polarization behavior of Phalloidin AF568 in cardiac muscle cells is similar to that in U2-OS cells. Although birefringence does exist in tissue level, it is out of scope for this manuscript because this work focuses on imaging subcellular structures within a single cell.

Reviewer #2

The authors provide a smart approach for combining the Nile Red emission wavelength sensitivity on the polarity of membranes with SIM super-resolution and polarization fluorescence microscopy. By combining the two parameters polarization modulation depth and spectral emission changes, they can distinguish many different membrane types with a single dye. That method is smart and should be published.

Response: We thank the reviewer for the high appreciation of our work.

Q1

I recommend that already in the abstract they better clarify how exactly they distinguish the ten different types of membranes, namely through the combination of these two parameters (Figure 3 e). Currently, the reader does not get from the statement in the abstract " This enables the high-throughput study of lipid heterogeneities of ten subcellular compartments, at different developmental stages, and even within the same organelle." what and how exactly the authors differentiate.

Response: Thank you for your suggestions. “High-throughput study” refers to optical properties acquired by super-resolution, multi-dimensional imaging. Currently, the membrane types of ten subcellular compartments cannot be fully distinguished only from the emission ratio and polarization modulation depth; therefore further morphology information has to be employed. Some types of compartmental membranes can be easily determined from their morphology, such as plasma membrane, nuclear membrane, and tunneling nanotubes. Yet, most organelles are round-shaped and with similar size, which cannot be distinguished from intensity images. Among these organelles, some of their types are determined when their polarity and phase were exclusively within one solid-line ellipse in Fig. 3e. However, there are some compartmental membranes whose lipid properties are within the overlapping area or out of any solid-line ellipse, so that their types cannot be determined. In future work, we plan to integrate artificial intelligence to achieve the automatic segmentation of different organelles.

To avoid confusion, we rephrased the sentence and added more details on how the organelles are determined.

Revisions are made in Line 5, Page 2: from

Simply using one dye that universally stains the lipid membranes, SPOT can simultaneously resolve the membrane morphology, polarity, and phase from the three optical-dimensions of intensity, spectrum, and polarization, respectively. These high-throughput optical properties reveal lipid heterogeneities of ten subcellular compartments, at different developmental stages, and even within the same organelle.

Q2

The study is also technically convincing. When reading the manuscript, it becomes convincingly clear how the authors can differentiate the different organelles based on the combination of Nile red polarization modulation depth and spectral emission changes.

However, although the authors well acknowledge the literature on SIM and Nile red, literature on fluorescence polarization microscopy is almost not cited, although their method depends largely on fluorescence polarization microscopy. They should better acknowledge which groundwork previously existed and which are the new parts of their approach.

Response: We thank the reviewer very much for this suggestion. There has been much research working on fluorescence polarization microscopy, which should be acknowledged. We have referenced existing polarization imaging techniques and compared them with SPOT.

Added discussion in Line 35, Page 12:

Compared with the other existing fluorescence polarization microscopy techniques, SPOT is superior in optical throughput that correlatively obtains the high-dimensional information from six raw images within tens of milliseconds. In contrast, the typical acquisition time for polarization modulation by point-scanning confocal imaging is in the range of seconds to minutes¹⁰. Polarization modulation by spinning disk confocal imaging allows parallel acquisitions that significantly increase the imaging speed to ~10 frames per second²⁷, but it is still slower than SPOT and has not become available with commercial systems. Single molecule localization microscopy (SMLM) is capable of measuring the orientation and wobbling of individual dipoles²⁸⁻³², and Nile Red, when using at ultra-low concentrations, is also compatible with SMLM³³. However, a higher concentration of Nile Red is required to stain all types of membranes, which rules out the use of SMLM imaging.

Q3

The authors also provide interesting examples of how their signals change during cell division. However, the extent to which new biophysical / biological insights are provided through this must be evaluated by reviewers who are experts in cell division.

Response: Thank you for your comments. Emerging research has begun to dissect the importance

of lipids in cell division, and previous studies have shown that lipid components are rearranged during division (see added Ref 4, 21, 22). The enrichment of several lipids at the cytokinetic furrow of dividing cells has been discovered, including cholesterol, sphingomyelin (SM) and

phosphatidylethanolamine (PE). This discovery may explain the modulation depth (lipid order) rises of plasma membrane during cell division in Fig. 4b.

Q4

Page 4 line 74: What do the authors mean by enormous in " Membranes in mammalian cells consist of enormous types of lipids"?

Response: Sorry for the expression. We rephrased the sentence to “Membranes in mammalian cells consist of a large variety of lipids”. By the way, the sentence was moved to Line 6, Page 4.

Q5

Page 10 line 205: “In this paper, we manually identify plasma membrane, nuclear membrane, TNT, ER, and mitochondria” What do the authors mean by manually?

Response: Sorry for bringing the confusion. What we want to express here is to identify these organelles through their morphological differences. We added a more detailed description of how the type of organelles was determined.

Revised text in Line 28, Page 13:

The lipid heterogeneity analysis offered by the polarity-phase plot of SPOT imaging opens the opportunities for organelle identification and segmentation. The synergistic use of membrane morphology, lipid polarity, and lipid phase can classify more than ten types of organelles. While plasma membrane, nuclear membrane, TNT, ER, and mitochondria with cristae structures are easily identified from their morphologies, other round-shaped organelles are inferred from their lipid properties. Artificial intelligence³⁵ can be further employed to integrate the intensity, polarity, and phase images for the automatic segmentation of the compartments in the future.

Q6

In summary, I recommend publishing when it is clear which new biophysical / biological insight about cell division are gained and after previous work about SIM, Nile red and fluorescence polarization microscopy is recognized in a more balanced way.

Response: We thank the reviewer very much for the support of our work.

Reviewer #3

In this manuscript, Zhanghao et al. obtained fine image of emission and polarization modulation of Nile red using SPOT technique.

- The data are interesting. However, the authors have oversimplified view of physical properties of lipids. Their results clearly show that there is no correlation between cholesterol/sphingolipid content and emission ratio/modulation depth.

Response: We are sorry for bringing the confusion of the correlation between the lipid composition and the measured lipid properties. Firstly, the emission ratio measured in our manuscript is consistent with previous results: sphingolipid or cholesterol has lower polarity and leads to a lower emission ratio (Ref 19 and 23). Furthermore, our results provided more details about different organelles with GFP colocalization. The correlation between measured emission ratio and the lipid composition of different organelle membranes is included in Supplementary Fig. 5. From these results, we can conclude that the emission ratio is directly related to the lipid polarity. We also add measurements of the lipid polarity in referenced solvents and in compartmental membranes (Supplementary Fig. 7).

Secondly, previous studies find that a higher lipid polarity led to a lower lipid ordered or the other way around, which is consistent with our results for mitochondria, late endosome, lysosome, early endosome, and plasma membrane. However, lipid diversity is much more complex in mammalian cells, and the lipid phase is also related to lipid-lipid interaction. These lead to counterexamples of lipid droplet and ER (the nuclear membrane is similar to ER membrane).

We have added more discussion of the correlation between the lipid properties and measured optical properties.

Added texts in Line 3, Page 4:

Two distinct physical properties of Nile Red are first explored in our work: the emission spectrum changes are measured by two-color ratiometric imaging (Fig. 2b), and the wobbling dynamics of fluorescent dipoles are resolved by polarization modulation (Fig. 2c). The emission spectrum of Nile Red will red shift in polar environment and blue shift in nonpolar environment⁶. Membranes in mammalian cells consist of three lipid categories: glycerophospholipid, sphingolipid, and cholesterol (Fig. 2d, e)². The lipid polarity descends from unsaturated

glycerophospholipid, saturated glycerophospholipid, sphingolipid, to cholesterol, so that the emission spectrum of Nile Red will shift from red to blue accordingly. The sphingolipid-rich membrane tends to form the solid-like phase, the membrane rich of cholesterol and saturated glycerophospholipid tends to form the liquid-ordered (Lo) phase, and the membrane rich of unsaturated glycerophospholipid tends to form the liquid-disordered (Ld) phase^{1,7}. Polarization imaging of fluorophores can reveal the orientational wobbling of molecular dipoles to distinguish ordered or disordered lipid phases⁸⁻¹¹. In the ordered membrane, the wobbling of dipoles is more restricted so that Nile Red exhibits stronger polarization response with increased modulation depth. In disordered membrane, the wobbling of dipoles is more flexible so that Nile Red exhibits weaker polarization response with decreased polarization modulation depth.

Added texts in Line 18, Page 12:

Previous studies only measured either the lipid polarity^{19,23,24} or the lipid phase⁸⁻¹¹. Some of these studies assumed that a higher lipid polarity led to a lower degree in lipid order or the other way around. This assumption is correct in the synthesized vesicles with one or two types of lipids. However, lipid diversity is much more complex in mammalian cells that produce more than 10,000 different types of lipids. Besides, the lipid phase is strongly associated with the lipid-lipid interactions², so that it should not be solely determined by the lipid polarity. Using SPOT, we presented for the first time the capability of correlative imaging of both the emission ratio and modulation depth, to quantify the complementary information of lipid polarity and phase. In our results, as the lipid polarity drops, the lipid phase increases in mitochondria, late endosome, lysosome, early endosome, and plasma membrane, which is consistent with the above assumption. The lower emission ratio suggests a higher proportion of sphingolipid or cholesterol that results in a higher lipid order. However, lipid droplet serves as a counterexample that shows both low lipid polarity and order. This is because the core of lipid droplets stores the low-polarity lipid, such as triglyceride in disorder. Nuclear membrane and ER have high lipid polarity, but with a relatively high ordered phase, which may be related to the sophisticated lipid-lipid interactions.

- Extensive lipidomics has to be accompanied to reach “a new horizon in correlating single-cell super-resolution lipidomics with multiplexed imaging of organelle interactome”.

Response: We thank the reviewer for the careful assessment of our work. We have rephrased the corresponding sentence.

Changes in Line 11, Page 2: from

This work suggests a new horizon in correlating single-cell super-resolution lipidomics with multiplexed imaging of organelle interactome.

Changed to:

This work suggests new research frontiers in correlating single-cell super-resolution lipidomics with multiplexed imaging of organelle interactome.

Q1

Fig. 3.

-Distribution of each organelle marker has to be shown.

Response: We thank the reviewer for this suggestion. We have revised Fig. 3 accordingly and have added markers on every statistical diagram.

Revised Fig. 3:

Figure 3 – Heterogeneity analysis of subcellular lipid membranes. (a, b) The polarity map and the phase map obtained by SPOT show noticeable contrasts between the lipid membranes of different compartments. The lipid polarity is quantified by the emission ratio and warm-color coded, while the lipid phase is resolved by the polarization modulation depth and cool-color

coded. (c-e) The statistics ($n \geq 24$ different organelles) measured on each colocalized compartment further quantify their lipid heterogeneity. The histogram of the emission ratio (c) shows the heterogeneous lipid polarity, and the histogram of the modulation depth (d) shows the heterogeneous lipid phase, where characters on the bars indicate significant differences. The polarity-phase plot (e) further categorizes the six groups of the compartments according to their similarities and differences, in which the solid line and the transparent ellipse show the standard deviation σ and 2σ of the measurement. (f, g) The polarity map and the phase map of mitochondria reveal the heterogeneity between the outer membrane and the cristae. The statistics ($n=24$ mitochondria) also show a significantly higher lipid polarity in the outer mitochondria membrane but no significant difference in the lipid phase. (h, i) The statistics ($n=24$) measured with colocalization show the increase in polarity and the decrease in phase from early endosomes to late endosomes. The arrows in (a, b) indicate the possible compartments recognized by their morphology and lipid properties. The statistical results are based on ≥ 3 independent experiments. EE: early endosome, G: Golgi apparatus, L: lysosome or late endosome, LD: lipid droplet, M: mitochondria, N: nuclear membrane, P: plasma membrane. Scale bar: (a-b) 10 μm ; (f-g) 2 μm .

-In this resolution, it is not clear whether the obtained difference in mitochondria is due to the cristae and other membranes or the heterogeneous lipid distribution in mitochondria.

Response: We thank the reviewer for pointing out this. We observed a significant difference in lipid polarity between the cristae and the outer membrane of mitochondria. The measurements were performed on 24 mitochondria, and this result is consistent with the lipid composition of mitochondria in Ref. 16 in the manuscript. Besides Fig. 3f, the lipid difference between the cristae and the outer membrane can also be observed in Fig. 4 and the supplementary movies. We have added this point to the manuscript.

Figure R4 – The lipid difference between the cristae and the outer membrane.

Added texts in Line 19 Page 7:

We observed the lipid heterogeneity on the outer membrane and the cristae of mitochondria directly from the functional images, and the statistics showed a significantly higher lipid polarity

in the cristae than the outer membrane (Fig. 3f, g). This observation can be further confirmed in the mitochondria images in Fig. 4 and Supplementary Movie.

-Late endosomes are multivesicular organelle with different lipid composition. Does their method distinguish heterogeneity of the organelle?

Response: Our technique has a resolution of 100-200 nm and only obtains an averaged measurement for features smaller than this size. Therefore, our method cannot clearly distinguish the lipid heterogeneity between the multiple vesicles within one late endosome. We have also added a discussion on the resolution limit of SPOT.

Added discussion in Line 9, Page 13:

Limitations of SPOT. Besides the optical sectioning in the axial dimension, the lateral resolution remains diffraction-limited for SPOT, while the resolution of SPOT-SIM3D is doubled. As a result, the polarity measurement is the averaged value of the lipid assembly within the point spread function (PSF). When curved membranous structures are within the PSF, the measurement will result in a significantly lower polarization modulation depth due to an ensemble dipole of the different molecular dipole orientations. For example, the membranous structure of Golgi apparatus is crowded and cannot be clearly resolved by SPOT, which further leads to a reduced lipid order in the measurement of Golgi apparatus.

-Why emission ratio is higher but modulation depth is lower in LE?

Response: During maturation from early endosomes to late endosomes, the proportion of cholesterol and sphingolipid is decreasing. Because the sphingolipid and cholesterol have low polarity and tend to form ordered membrane, the lipid polarity increases in LE and the lipid order decreases (resulting in a lower modulation depth).

-Does their method detect heterogeneity of plasma membrane?

Response: Yes, SPOT can detect the heterogeneity, as shown in Fig. 4b. We could see the heterogeneity and dynamic changes of the plasma membrane during cell division from Fig. 4b. The lipid polarity (emission ratio) dropped, and the lipid order (modulation depth) rose in the plasma membrane at the division location, while the non-dividing plasma membranes remain unchanged. However, SPOT cannot detect the lipid rafts with a size smaller than 100 nm.

Figure R5 –The heterogeneity of the plasma membrane. The emission ratio drops during the division and recovers after the separation of two plasma membranes, while the modulation depth first rises and then drops. During the same period, the control plasma membrane that is not dividing keeps constant in both polarity and phase.

-What are alphabets in Fig. 3c and d?

Response: The alphabets indicate the significant difference between the measurements, and we have added this description in the figure caption.

-“During maturation, late endosomes lose cholesterol and sphingolipid” Does this mean degradation of sphingolipid in late endosomes and cholesterol exit by NPC1? Phospholipids are also degraded in late endosomes/lysosomes.

Response: From Ref. 19 in the manuscript, we could see that the degradation of cholesterol of late endosome is due to the extraction by NPC1 and NPC2 proteins. Cholesterol extraction from late endosomes is also associated with sphingomyelin hydrolysis into ceramide. Based on our results, it seems that the sphingolipid and cholesterol are degraded faster than phospholipids. Here we just focus on the changes in the relative lipid composition contents. Since we are not experts in the endosome research, we could not infer further conclusions, such as how phospholipids may change during maturation.

Q2

Fig. 4.

-What are alphabets in Fig. 4c?

Response: The alphabets indicate the significant difference between the measurements, and we have added this description in the figure caption (Line 9, Page 12). The histogram plots the pixel values on each TNT. Variance among multiple groups are analyzed by one-way ANOVA ($p < 0.05$) in SPSS, and the characters on the bars indicate significant differences of each group.

-Detailed explanation of the methods is required. How reproducible are these results?

Response: We thank the reviewer for this important suggestion. We added the details of SPOT reconstruction process in Supplementary Note.

For data repeatability: the measurements on compartmental membranes are performed on ≥ 24 different compartments imaged in ≥ 3 independent experiments; the measurements of membrane rearrangements during cell division calculate pixel statistics and are repeated in ≥ 2 independent experiments. We also validate our experiments to avoid the measurement errors induced by channel registration, motion blur, or photobleaching, which is included in Supplementary Note.

Added description of data validation in Supplementary Note (please see the detailed explanation of SPOT reconstruction process in the Supplementary Information):

g) Validation of reconstruction results

We also take the following steps to further check the processed data to improve the reliability of the measurement:

- i. Double-check the channel registration. Arrange different colors for the registered images of different detection channels and get a composite image. Observe the nuclear membrane part of the composite image. If the nuclear membranes of different channels are misaligned, go back to step a) and remeasure the affine transformation matrix.
- ii. Check the fluctuation of modulation depth. This step is mainly used to exclude the data that motion blur or photobleaching have a significant impact on the measurement. As shown in step f), at least 3 polarization modulations are required to get modulation depth OUF and dipole orientation β , and we adopted $\theta_i = 0^\circ, 60^\circ, 120^\circ$ in the experiment. For the data check, we carry out 5 polarization modulations in sequence with $\theta_i = 0^\circ, 60^\circ, 120^\circ, 0^\circ, 60^\circ$. When selecting $\theta_i = 0^\circ, 60^\circ, 120^\circ$, we can find the OUF. Similarly, when choosing $\theta_i = 60^\circ, 120^\circ, 0^\circ$ and $\theta_i = 120^\circ, 0^\circ, 60^\circ$, we can also solve the modulation depth, expressed as OUF' and OUF''. OUF, OUF' and OUF'' should be

close to each other if the motion blur and photobleaching are not obvious. We excluded the data where OUF, OUF' and OUF'' differ more than 15%.”

Reviewers' Comments:

Reviewer #1:

Remarks to the Author:

The revised manuscript is significantly improved over the original version, including much more detailed information on the experimental side, simulations accounting for noise, and addressing accuracy issues. There is one remaining weakness of the approach which is that the polarized sensitive data are strongly dependent on the nanoscale membrane morphology, which can change from one membrane type to another. This limits strongly the data interpretation, however this is nevertheless properly pointed out by the authors. Overall, I consider that this version is suitable for publication.

Reviewer #2:

None

Reviewer #3:

Remarks to the Author:

Nile red does not directly see lipids, rather it senses polarity and hydration of microenvironment. Water content and protein density are different in different organelle. Thus, the observed Nile red signal is not only affected by lipids. The authors need to clearly mention this.

It is still not clear how did they identify G in Fig. 3a as Golgi and L as late endosomes etc. The photos of organelle markers have to be shown together with Fig. 3a and b.

Fig. 2 (c): The color of the figure is not consistent with the description in Fig legend.

Fig. 4(d). Figure (TNT) and legend (mitochondria) are not consistent.

Fig. 4 (e). The structures of a-c are dynamically changed. How did they calculate the values of the graph? How reliable is the obtained value?

Response to Reviewers' Comments

We very much appreciate the critical reading of our manuscript and valuable suggestions by the reviewers. We have carefully reviewed the comments and revised the manuscript accordingly. The revised text is underscored in the manuscript and is also included here. The responses to the comments are listed point-by-point as follows in blue (the page and line numbers refer to the revised manuscript):

One of the reviewers mentioned doi: 10.1038/nmeth.2919 as a relevant work that should be cited and discussed.

Response: We have cited and discussed the relevant work in the revised manuscript.

Added texts in Line 5, Page 13:

Polarization demodulation can also obtain super resolution with deconvolution in spatioangular space²⁸⁻³⁰. However, this technique requires a sparse distribution of dipole orientations that is not the case in cells with crowded membranous structures.

Reviewer #1

The revised manuscript is significantly improved over the original version, including much more detailed information on the experimental side, simulations accounting for noise, and addressing accuracy issues. There is one remaining weakness of the approach which is that the polarized sensitive data are strongly dependent on the nanoscale membrane morphology, which can change from one membrane type to another. This limits strongly the data interpretation, however this is nevertheless properly pointed out by the authors. Overall, I consider that this version is suitable for publication.

Response: We thank the reviewer for the appreciation of our revised manuscript. The suggestions significantly improved the quality of our manuscript.

Reviewer #3

Q1

Nile red does not directly see lipids, rather it senses polarity and hydration of microenvironment. Water content and protein density are different in different organelle. Thus, the observed Nile red signal is not only affected by lipids. The authors need to clearly mention this.

Response: We thank the reviewer for these comments. It is true that the emission spectrum of Nile red senses the polarity of microenvironment. When Nile red is in water or lipoproteins, its emission spectrum is also shifted. However, Nile Red is hardly soluble in water and the partition coefficients of Nile red in organic solvents relative to water are ~200 (Greenspan, P., Mayer, E. P. & Fowler, S. D. The Journal of Cell Biology 100, 965-973 (1985), doi:10.1083/JCB.100.3.965.). Besides, the fluorescence intensity of Nile red in polar solvents are much weaker than in nonpolar solvents. For example, the relative fluorescence intensity in *n*-Heptane (nonpolar) is ~2000-fold greater than in ethanol (polar). The surface modification proteins of lipid such as lipoprotein will not influence the spectroscopy distribution of Nile Red. (See reference: Gaus, K. et al. Proceedings of the National Academy of Sciences 100, 15554, (2003). Therefore, we suppose that the water content and protein density only have slight influence on the emission spectrum of Nile red. We have added relative discussion in our manuscript.

Added texts in Line 26, Page 13:

The spectrum shift is mostly attributed to the polarity of lipid, because the fluorescence of Nile Red is negligible in water and the emission spectrum of lipid dyes is little affected by lipid surface modification protein such as lipoprotein⁷.

Q2

It is still not clear how did they identify G in Fig. 3a as Golgi and L as late endosomes etc. The photos of organelle markers have to be shown together with Fig. 3a and b.

Response: We thank the reviewer for the suggestion. The organelles in Fig. 3-4 were not identified from other colocalized organelle markers. Instead, we identify the organelle types comprehensively from their morphologies and lipid properties, which is illustrated below in detail.

Fig. Identification of membrane types via the emission ratio and polarization modulation depth. With SPOT imaging, we obtain three images of the intensity, emission ratio, and polarization modulation depth. The intensity image of Fig. 3a (emission ratio) and Fig. 3b (modulation depth) is shown in (a). For each pixel, it will be marked with a specific type if its emission ratio and modulation depth fall within the solid-line ellipse in Fig. 3e of the corresponding type. Taking lipid droplets for example, the mean and standard deviation of emission ratio are $ave_{ld,polarity} = 0.2191$ and $std_{ld,polarity} = 0.07269$, and the mean and standard deviation of modulation depth are $ave_{ld,phase} = 0.3512$ and $std_{ld,phase} = 0.1047$. Then the pixel will be marked as the type of lipid droplet if its emission ratio is within $[ave_{ld,polarity} - 2std_{ld,polarity}, ave_{ld,polarity} + 2std_{ld,polarity}]$ and its modulation depth is within $[ave_{ld,phase} - 2std_{ld,phase}, ave_{ld,phase} + 2std_{ld,phase}]$. If one pixel is simultaneously marked as more than one types, the pixel will be further assigned with no type. For pixels that are marked with a specific type, it will be displayed in the same color with the up-right texts. (b,c) The zoom-in images in the rectangle areas in (a). We manually decide the type of a compartment if most pixels within the compartment are with the same type. For plasma membrane, nuclear membrane, ER, and mitochondria with cristae structures, we identify their types from their morphologies. This manual method of segmentation is low in efficiency. It also fails to identify the types of many compartments because of the large standard deviations. We plan to employ deep learning in our future work to utilize both the information of morphology and membrane properties. Scale bar: (a) 10 μm ; (b-c) 2 μm .

Q3

Fig. 2 (c): The color of the figure is not consistent with the description in Fig legend.

Response: We thank the reviewer for the careful assessment of our work. We have corrected the corresponding figure legend.

Revised Fig.2(c) legend in Line 7, Page 6:

(c) When Nile Red inserts into the lipid membrane, the wobbling behavior of the fluorescent dipole reflects the lipid phase, which can be quantified by polarization modulation depth. In the ordered phase, the dipole orientation is more uniform, leading to a higher modulation depth (the schematic curve in blue); while in the disordered phase, the dipole orientation is more anomalous, resulting in a smaller modulation depth (the schematic curve in red).

Q4

Fig. 4(d). Figure (TNT) and legend (mitochondria) are not consistent.

Response: We thank the reviewer for the careful assessment of our work. The legend of Fig. 4(d) has been corrected.

Revised Fig. 4 legend in Line 1, Page 12:

Figure 4 – Long-term monitoring of the lipid dynamics of multiple compartments. (a) The time-lapse images recorded the late-stage division of two U2-OS cells at 3 s acquisition interval for 10 mins. The bleaching analysis in (f) shows the non-bleaching imaging power at this interval. (b) The enlarged images of the yellow box show the lipid polarity and phase during the division of the plasma membrane. The emission ratio drops during the division and recovers after the separation of two plasma membranes, while the modulation depth first rises and then drops. During the same period, the control plasma membrane that is not dividing keeps constant in both polarity and phase. (c,d) The enlarged images of the purple box show the formation of TNTs after the separation of plasma membranes. The measurements reveal a large variation in the lipid polarity and phase among these TNTs. The histogram plots the pixel values on each TNT, and the characters on the bars indicate significant differences of each group. (e) The enlarged images of the blue box show the cristae dissociation of mitochondria during the process. The curves of Mito-b and Mito-c show an increase in the outer membrane in the emission ratio, which drops back afterward. In contrast, the lipid polarity of Mito-1 without cristae dissociation is more constant. (f) The curves show the average fluorescence signal imaged with different acquisition interval and are fitted with an exponential function to calculate the half-time. The exposure time of each image is 40 ms, and the total acquisition time of a SPOT dataset (six raw images) is 240 ms. Similar results are observed in ≥ 2 experiments. Scale bar: (a) 5 μm ; (b-g) 2 μm .

Q5

Fig. 4 (e). The structures of a-c are dynamically changed. How did they calculate the values of the graph? How reliable is the obtained value?

Response: In time series images, we manually marked the position of the selected mitochondria for each image, as shown in the Fig. R2(b). Then we analyzed the statistics of the selected area. The dynamic changes of the mitochondrial morphology are continuous and can be easily recognized visually. We also selected a segment of plasma membrane that did not participate in the division as a control. The stable emission ratio and modulation depth of the selected plasma membrane from 15 to 90 s (Fig. R2(c)) also proves the reliability of our manual labeling.

Fig. R2 – The manually labeled position of the lipid membranes and their dynamic changes.

(a) Optical sectioning image at 0 s. (b) Manually labeled ROI curve of the plasma membrane (figures in the top two rows) and mitochondrial outer membrane (figures in the bottom two rows). Polyines of the same color track the position of the same structure. The red, orange and magenta lines correspond to 3 different mitochondria, denoted as Mito-a, Mito-b and Mito-c. Cristae

dissociation occurs in Mito-b and Mito-c, while Mito-a does not. (c) The dynamic changes of the emission ratio and modulation depth in the selected structures.